# Stress during pubertal development affects female sociosexual behavior in mice

**Yassine Bentefour** [1] ✉ **& Julie Bakker** [1] ✉

Puberty is a crucial phase for the development of female sexual behavior. Growing evidence suggests that stress during this period may interfere with the development of sexual behavior. However, the neural circuits involved in this alteration remain elusive. Here, we demonstrated in mice that pubertal stress permanently disrupted sexual performance without affecting sexual preference. This was associated with a reduced expression and activation of neuronal nitric oxide synthase (nNOS) in the ventrolateral part of the ventromedial hypothalamus (VMHvl). Fiber photometry revealed that VMHvl nNOS neurons are strongly responsive to male olfactory cues with this activation being substantially reduced in pubertally stressed females. Finally, treatment with a NO donor partially restored sexual performance in pubertally stressed females. This study provides insights into the involvement of VMHvl nNOS in the processing of olfactory cues important for the expression of female sexual behavior. In addition, exposure to stress during puberty disrupts the integration of male olfactory cues leading to reduced sexual behavior.

Low sexual desire is a deleterious condition that causes marked distress and interpersonal difficulties. It has a general negative impact on the quality of life. The prevalence of low sexual desire is high: up to 39.5% of women aged between 18 and 44 years old reported deficient or absent sexual fantasies, sexual arousal, and orgasm[1]. Many studies have proposed a potential link between testosterone and androgen metabolite levels and their capability to promote sexual desire[2–4], but no clear association has been reported. Growing evidence suggests that exposure to stress during pubertal development can be associated with permanent disruption of female reproductive behavior[5–8]. Puberty is defined as the transition to a mature reproductive state. In mice, it starts with the hormonal changes that lead to the onset of the first external sign of pubertal development, vaginal opening, and ends with the onset of the first reproductive cycle[9,10]. Indicating that pubertal development can take weeks and can potentially be influenced by stress or other factors[11,12]. For instance, it has been reported that transport stress or an immune challenge at the age of 6 weeks induces a decreased behavioral response to steroid hormones during adulthood[7], indicating the importance of this period in the development of female sexual behavior. Even though the gestational period has been considered as an important phase for the organization of reproductive behavior, the pubertal stage has also been found to be critical for the development of

the female brain and as consequence sexual behavior[13,14]. For example, the study by Gerall and colleagues in which they performed ovariectomies in females at different postnatal ages have suggested the importance of ovarian hormones around puberty for the normal development of sexual receptivity[15]. Moreover, aromatase knockout (ArKO) females that are deficient in estradiol, displayed a strongly reduced sexual behavior in adulthood[13], which could be restored by estradiol supplementation from postnatal days 15 to 25[14]. Finally, female mice exposed to prolonged social isolation from P25 to P60 showed reduced levels of female sexual behavior compared to group-housed females[16]. Indeed, chronic social isolation is a robust stressor that is known to induce multiple long-lasting effects especially in social species such as rodents and humans. For instance, it has been shown that exposure of female prairie voles to two months of social isolation is associated with behavioral alterations related to stress-induced endocrine responses, including anhedonia, increased circulating levels of corticosterone and elevated number of cells coexpressing Fos and corticotropin-releasing factor in the paraventricular nucleus (PVN)[17]. Corticosterone-related plasticity has been observed in several brain regions of the hypothalamus that play an important role in the regulation of sexual behavior. Exposure to social isolation during puberty was reported to disrupt the distribution of estrogen receptor-α in the

¹GIGA Neurosciences-Neuroendocrinology Lab – University of Liège, Liège 4000, Belgium. ✉e-mail: ybentefour@uliege.be; jbakker@uliege.be

medial preoptic area (MPOA) and bed nucleus of the stria terminalis (BST) in prairie voles[18], the anteroventral periventricular nucleus (AVPV) and the ventromedial nucleus of the hypothalamus (VMH) in mice[16].

Recently, it has been shown that kisspeptin neurons located in the rostral periventricular area of the third ventricle (RP3V)[19] and nNOS neurons in the ventrolateral part of the ventromedial hypothalamus (VMHvl)[20,21] are two specialized neuronal populations necessary for the expression and the modulation of female sexual behavior, with nNOS neurons being a downstream target of kisspeptin. Interestingly, the RP3V kisspeptin neuronal population develops during puberty under the influence of estradiol[22]. Indeed, it has been reported that the number of RP3V kisspeptin neurons is strongly reduced in adult ArKO female mice, whereas postnatal treatment with estradiol increased the number of kisspeptin-expressing neurons[22,23]. Therefore, exposure to stress during puberty might interfere with the normal development of this neural circuit modulating female sexual behavior.

Likewise, elevated stress hormones can negatively regulate the reproductive axis (reviewed in ref. 24). For instance, recent clinical studies have linked dehydroepiandrosterone (DHEA) to sexual desire[2,4]. DHEA is a prohormone to estradiol and one of the most abundant steroids that is produced by the hypothalamic-pituitary-adrenal (HPA) axis along with cortisol in response to stress. Therefore, a disruption of HPA functioning might lead to a disruption of sex hormones[25,26] and thus influence the neural development of female sexual behavior. It is known that high levels of chronic stress, primary during childhood, can lead to the dysregulation of the HPA axis, resulting in abnormal secretion of stress hormones. For example, studies of people with a history of childhood sexual abuse report abnormal HPA activity, including low morning cortisol concentrations and a blunted cortisol diurnal slope[27]. Similar observations were reported in adult individuals exposed to early life stress including physical abuse[28] and maternal stress[29]. Interestingly, women diagnosed with low sexual desire were reported to have low cortisol and DHEA levels[8]. Thus, we hypothesize that sexual dysfunction could be the result of HPA dysregulation induced by exposure to stress during pubertal development. Therefore, we investigated the effects of chronic stress over the pubertal period on female sexual functioning and the neuronal circuit regulating its expression.

## Results

### Pubertal stress disrupts the estrous cycle and affects sexual behavior in female mice

The first external sign of sexual maturation is vaginal opening. We found that pubertal stress by social isolation had no effect on the age of vaginal opening (Fig. 1a, b). However, adult female mice that were exposed to stress during puberty showed a substantial disruption of the estrous cycle, spending less time in proestrus ($p = 0.0009$) and estrus ($p = 0.0041$) and more time in metestrus ($p = 0.0003$) compared to the control non-stressed females (Fig. 1c–f). Pubertally stressed females also displayed significantly fewer cycles ($p = 0.0029$) during the three weeks of vaginal sampling (Fig. 1g). Furthermore, females exposed to pubertal stress displayed less lordosis compared to the control non-stressed females ($p = 0.0333$) (Fig. 1h). Further examination of the lordosis data revealed that pubertally stressed females showed a bimodal distribution indicating the presence of two subpopulations that were different in their sexual receptivity. Subjects with an average lordosis quotient equal or below 30% were categorized as minimally receptive (MR) females, while highly receptive (HR) females were defined as subjects with an average lordosis quotient above 30%. According to these criteria, we observed that the percentage of MR females in the control group was around 11% (Fig. 1i), whereas it was 62% in the group exposed to pubertal stress (Fig. 1j). The observed low levels of lordosis behavior were not associated with a decreased attractiveness of the females or a lower performance of the stimulus males. The number of mounts performed by the males was similar between the two groups (repeated measures (RM) two-way ANOVA; group effect: $F_{(1,31)} = 1.780$; $p = 0.1919$) and consistent across all trials (RM two-way ANOVA; trial effect: $F_{(3,93)} = 1.780$; $p = 0.1564$) (Supplementary Fig. s1).

The sexual motivation of the females was also evaluated using the partner preference test. It was found that both groups of females displayed a significant preference for the male (control: $p = 0.0024$; pubertal stress: $p = 0.0143$) with no statistical difference between the control and the female mice exposed to pubertal stress ($p = 0.784$) (Fig. 1k). Moreover, no significant correlation was found between lordosis quotients and preference score ($r = −160$; $p = 0.373$) (Fig. 1l), indicating that MR females like HR females showed a preference for the male.

Decreased sexual performance might be associated with an increased state of anxiety or depression-like behaviors. Therefore, we evaluated this using the elevated plus maze and the forced swim test. It was found that females exposed to pubertal stress spent similar time in the open arms of the elevated plus maze (Supplementary Fig. s2a, b) and a similar duration of resting immobile in the forced swim test as control non-stressed females (Supplementary Fig. s2c). Taken together, these data suggest that the reduced sexual performance following exposure to pubertal stress was not induced by an increased state of anxiety or depression-like behaviors.

### Pubertal stress has no effect on circulating levels of steroid hormones

Ovarian steroid hormones, including estradiol (E2) and progesterone (P4) are necessary for the development and the display of lordosis behavior. Thus, we assessed the levels of circulating E2 during puberty and adulthood. We found no effect of pubertal stress on the concentration of E2 at the age of P40 (Fig. 2a) as well as P60 (Fig. 2b). In addition, compared to the control group, no difference was observed in the levels of progesterone at the age of P60 (Fig. 2c) in the pubertally stressed females. Also, no correlation was observed between lordosis quotient and ovarian hormones (Supplementary Fig s3a–c).

Recent reports support the idea that the HPA axis could be involved in the permanent disruption of female sexual behavior[8,30]. The balance between corticosterone and DHEA release is a hallmark that reflects the overall functioning of the HPA axis[31]. Therefore, we collected blood samples in ovary intact adult females under two conditions; home cage to evaluate baseline levels of stress hormones, and 30 min following exposure to the elevated plus maze to assess the response of the HPA axis (Fig. 2d). Measurement of the circulating stress hormones in the control and in females exposed to pubertal stress revealed a significant increase of corticosterone (RM two-way ANOVA: treatment effect ($F_{(1,21)} = 139.4$; $p < 0.0001$) (Fig. 2e), and a significant drop in DHEA levels compared to the baseline concentrations in both groups (RM two-way ANOVA: treatment effect ($F_{(1,36)} = 22.84$; $p < 0.0001$)) (Fig. 2f). In addition, there was no significant difference between the control and pubertally stressed females in the concentrations of plasma corticosterone and DHEA at baseline levels and following stimulation of the HPA axis. Corticosterone/DHEA ratio was also not different between the two groups (Fig. 2g). In addition, no correlation was observed between lordosis behavior and corticosterone (Supplementary Fig. s3d, g) or DHEA (Supplementary Fig. s3e, h). However, we found a significant correlation between baseline CORT/DHEA ratio and female sexual receptivity ($r = 0.444$; $p = 0.049$) (Supplementary Fig s3f). Taken together, these data suggest that sexual dysfunction following pubertal stress is associated with baseline levels of circulating corticosterone and DHEA.

### Sexual dysfunction in females exposed to pubertal stress is linked to a decreased number and activation of nNOS neurons in the VMHvl

Recently, it has been found that female sexual behavior is orchestrated by the neuronal population of kisspeptin located in the RP3V[19] This

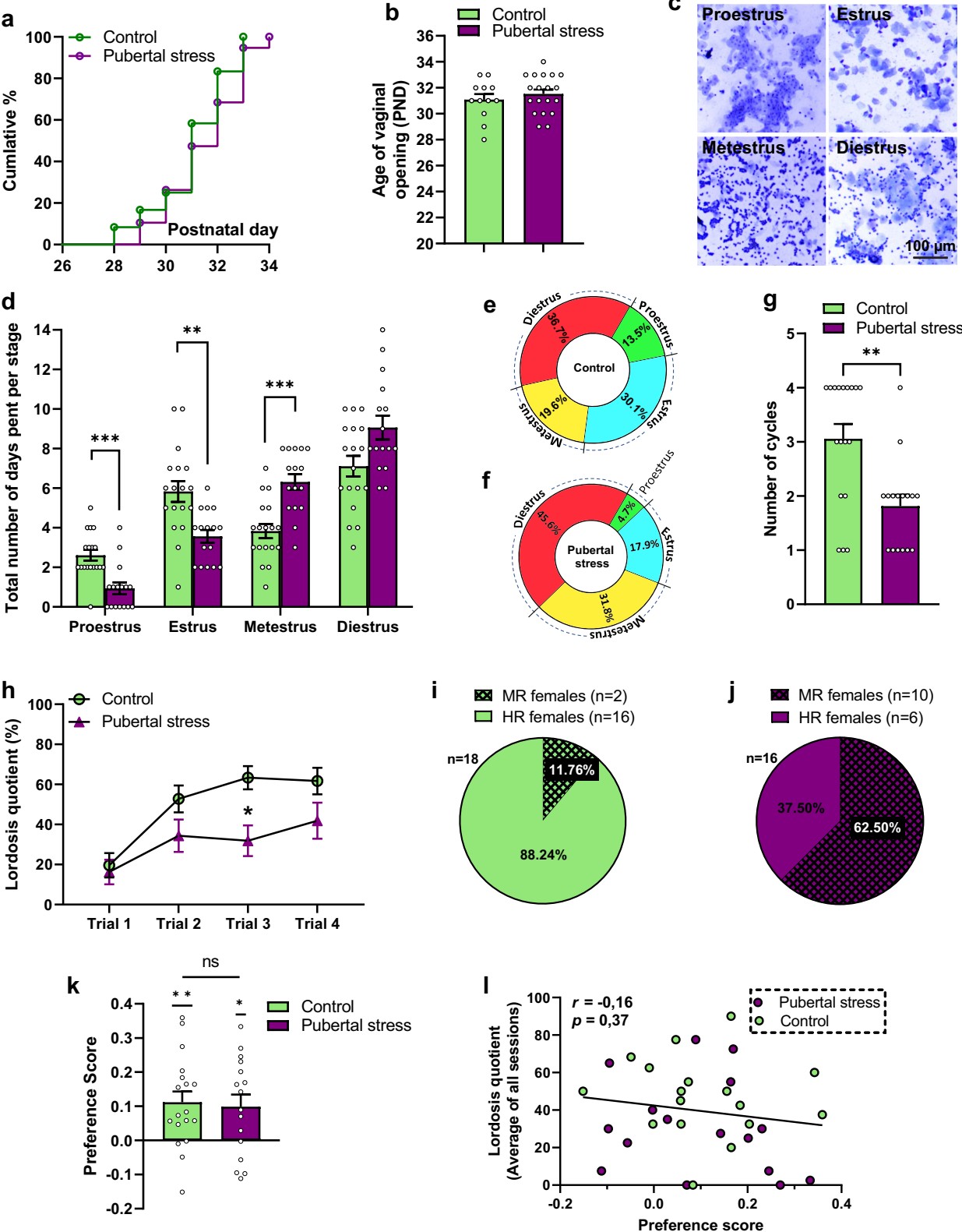

population is specifically activated by male olfactory cues[32]. In addition, it has been shown that kisspeptin neurons in the RP3V express both receptors for glucocorticoids and corticosterone-releasing factor[33]. Therefore, we hypothesized that kisspeptin neurons may be disrupted in female mice exposed to pubertal stress. Using immunohistochemical studies (Fig. 3a, b), we found that the number of neurons expressing kisspeptin in the RP3V increased in both groups following

exposure to male bedding ($F(1,19) = 4.968$; $p = 0.0381$), without any significant difference between the groups ($F(1,19) = 0.1769$; $p = 0.6787$). The interaction between the two factors was also not significant ($F(1,19) = 0.3907$; $p = 0.5394$) (Fig. 3c). Similar observations were found for the coexpression of Fos and kisspeptin in the RP3V. Following exposure to male bedding, no difference in the number of Fos-expressing kisspeptin neurons was observed between females

**Fig. 1 | Female mice exposed to pubertal stress display disrupted estrous cycle and decreased sexual performance. a** Cumulative percentage of females showing vaginal opening. **b** Age of vaginal opening in pubertally stressed females compared to control subjects. Two-tailed unpaired $t$ test ($t(29) = 0.8109$, $p = 0.424$). $n = 12$ for the control and $n = 19$ for the pubertal stress group. **c** Representative images of the different stages of the estrous cycle. **d** Total number of days spent in each stage of the estrous cycle in females exposed to pubertal stress versus control, non-stressed subjects. Two-way RM ANOVA followed by Bonferroni's multiple comparison; factor group: $F(1,32) = 1.231$, $p = 0.2755$; cycle stage: $F(2.304, 73.72) = 55.69$, $p < 0.0001$); group × cycle stage: $F(3, 96) = 12.43$; $p < 0.0001$). $**p = 0.0041$; $***p = 0.0009$ for post hoc comparison in proestrus and $p = 0.0003$ for metestrus. $n = 18$ for the control, $n = 16$ for pubertal stress group. **e, f** Pie charts of the average percentage of time spent in each stage of the estrous cycle in the control and pubertal stress group. **g** Number of estrous cycles that females went through during the 3-week period of the monitoring of the estrous cycle. Two-tailed

Mann–Whitney test: $U = 61.50$, $p = 0.0029$. $n = 18$ for the control, $n = 16$ for pubertal stress group. **h** Evolution of lordosis quotient in all trials. Two-way RM ANOVA followed by Bonferroni; factor group: $F(1, 32) = 4.950$, $p = 0.033$; session: $F(2.539,81.24) = 22.03$, $p < 0.0001$; interaction between the two factors: $F(3,96) = 3.152$, $p = 0.0285$). $*p = 0.011$. $n = 18$ for the control, $n = 16$ for pubertal stress group. **i** Percentage of highly (HR) and minimally (MR) receptive females in the control group. **j** Percentage of HR versus MR females in the group of female mice exposed to pubertal stress. **k** Preference score for the control and pubertal stress group. One-sample $t$ test (hypothetical value = 0) (control: $t(17) = 3.571$, $p = 0.0024$; Pubertal stress: $t(15) = 2.77$, $p = 0.0143$). ns: not significant; unpaired $t$ test (two-tailed) $t(32) = 0.2754$, $p = 0.784$. Control ($n = 18$), pubertal stress ($n = 16$). **l** No significant correlation between preference scores and lordosis quotients. Pearson's correlation $r = -0.16$, $p = 0.3738$. All data are presented as mean ± SEM. Corrections were made whenever multiple comparisons test is used. Source data are provided as a Source Data file.

exposed to pubertal stress and non-stressed females ($F(1,19) = 2.778$; $p = 0.1119$) (Fig. 3d), suggesting that the olfactory signal was transduced normally towards the RP3V kisspeptin neurons. Moreover, no significant correlations were found between lordosis quotients and the number or the activation of the RP3V kisspeptin neurons (Supplementary Fig. s4).

Further analysis of Fos expression revealed that female mice exposed to pubertal stress showed similar activation patterns in multiple nuclei involved in female sexual behavior in response to male bedding, except for the VMHvl. Following exposure to male bedding, a significant decrease was found in the number of Fos expressing neurons in the VMHvl of females exposed to pubertal stress (Fig. 3e, Supplementary Table s1).

Previously, it has been shown that nNOS expressing neurons in the VMHvl are an important downstream target of RP3V kisspeptin neurons in the neural circuit underlying lordosis behavior[19–21]. Therefore, we analyzed the number of VMHvl nNOS neurons as well as whether they were activated or not by male olfactory cues. Compared to the control, a strong decrease, around 40%, in the number of nNOS neurons located in the VMHvl was observed in females exposed to pubertal stress (Two-way ANOVA; group effect ($F(1,18) = 16.25$; $p = 0.0008$)). However, exposure to male bedding did not affect the number of nNOS neurons in either female group (Two-way ANOVA; clean vs male bedding ($F(1,18) = 0.068$; $p = 0.796$)) (Fig. 3e, f).

Analysis of VMHvl nNOS/Fos coexpression revealed that exposure to male bedding induced a significant increase in the number of nNOS neurons coexpressing Fos compared to clean bedding (Two-way ANOVA; bedding effect ($F(1,18) = 51.23$; $p < 0.0001$), with a significant difference between the groups (Two-way ANOVA; group effect ($F(1,18) = 4.684$; $p = 0.044$)) (Fig. 3g). The interaction between the two factors was also significant ($F(1,18) = 5.42$; $p = 0.032$). Post hoc analyses indicated no difference between control and females exposed to pubertal stress following exposure to clean bedding in the number of activated nNOS neurons ($p > 0.99$). However, upon exposure to male bedding, females exposed to pubertal stress displayed a significantly reduced number of neurons coexpressing nNOS and Fos in the VMHvl compared to the control group ($p = 0.015$). Indeed, females exposed to pubertal stress displayed an activation of only 15.6% of the VMHvl nNOS neurons compared to 24.8% in control females (Fig. 3g).

Further analyses revealed that the reduction in the number of VMHvl nNOS neurons was specific to the cohort of MR females following exposure to pubertal stress, whereas HR females expressed similar number of nNOS neurons compared to the control, non-stressed group (Supplementary Fig. s5a, b). Moreover, analysis of the PVN nNOS population showed no difference in the number of neurons expressing nNOS between the control and females exposed to pubertal stress (Kruskal–Wallis test; $H(3) = 4.386$; $p = 0.107$) (Supplementary Fig. s5c, d), suggesting that the reduced number of nNOS neurons following exposure to pubertal stress is specific to the VMHvl population.

Analysis of nNOS and glucocorticoid receptor (GR) coexpression in the VMHvl and PVN indicated that only around 10% of the nNOS neurons in both nuclei are expressing GR (Fig. 4). Further analysis revealed that, compared to the control group, no difference was found in the number of nNOS neurons in the VMHvl coexpressing GR in both HR and MR females (Fig. 4b). A similar observation was made for the PVN. No difference between the control subjects and females exposed to pubertal stress in the number of nNOS neurons colocalizing with GR (Fig. 4f).

## VMHvl nNOS neurons are specifically responsive to male olfactory cues

Next, using fiber photometry, we analyzed the activity of nNOS neurons during lordosis behavior. To do so, we selectively induced the expression of the calcium sensor (GCaMP6s) into the VMHvl of nNOS::Cre females (Fig. 5a, b). Viral transfection was established using a Cre recombinase-dependent AAV virus serotype 1 in which the GCaMP6 is under the control of the synapsin promoter (AAV1.Syn.-Flex.GCaMP6s.WPRE.SV40). This approach provided an average transfection rate of $70.23 ± 2.11\%$ of VMHvl nNOS neurons (Fig. 5c).

Data analysis revealed that introducing the stimulus male to the recording aquarium at the beginning of the lordosis test robustly increased the activity of nNOS neurons (Fig. 5d–g). Statistical analysis showed a significant decrease in nNOS activation in MR ($p = 0.0485$) but not HR ($p > 0.99$) females that were exposed to pubertal stress compared to control, non-stressed females (Fig. 5p).

Behavioral analyses confirmed the presence of two subpopulations of females following exposure to pubertal stress that are different in their levels of sexual receptiveness. MR females, which are characterized by expressing low percentage of lordosis quotient (Fig. 5n), also showed a high percentage of rejection quotients compared to the control group (Dunn's multiple compressions; $p = 0.0481$), whereas the HR females were not statistically different from the control non-stressed subjects ($p > 0.99$) (Fig. 5o).

Next, we recorded the activity of nNOS neurons when females expressed lordosis and rejection behaviors. Remarkably, VMHvl nNOS neurons showed very low levels of activation when females, from all experimental groups, expressed lordosis behavior (Fig. 5h–j, q). Measuring the activation of the VMHvl nNOS neurons when females rejected the stimulus males yielded similar results as observed for lordosis (Fig. 5k–m, r), suggesting that VMHvl nNOS neurons are not directly involved in the expression of lordosis or rejection behavior.

Since we observed a robust activation of nNOS neurons in response to the introduction of the stimulus male at the beginning of the lordosis test, we next analyzed whether these neurons are responsive to volatile olfactory cues. nNOS neurons activity was significantly increased in control, non-stressed females, in response to male urine (Fig. 6a, b) but not to female urine (Fig. 6c, d), water (Fig. 6g, h) or a non-social odor such as amyl-acetate (Fig. 6e, f), (multiple

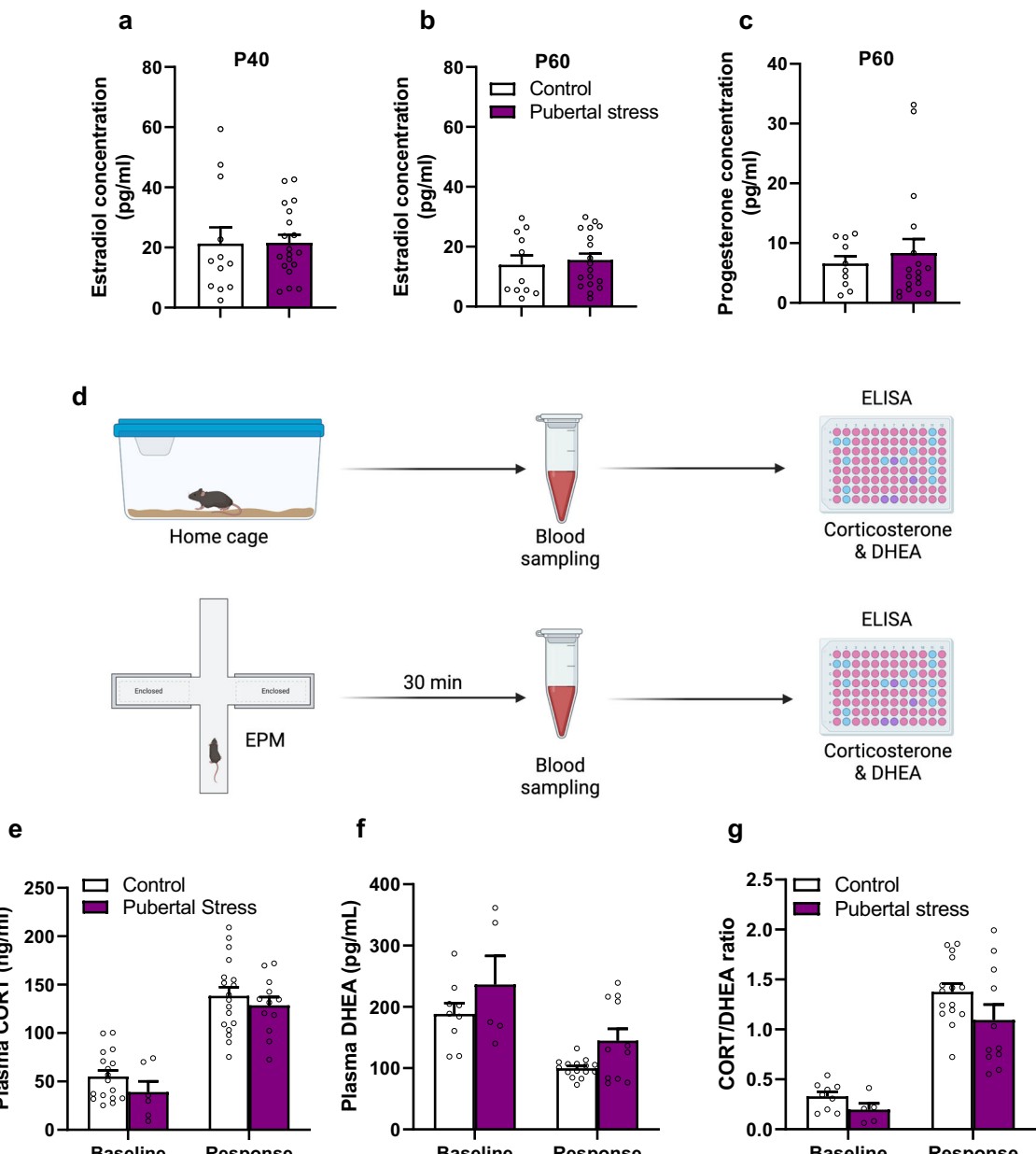

**Fig. 2 | Effect of pubertal stress on HPG and HPA axis. a** Estradiol concentrations in control and pubertally stressed females at the age of P40. Mann–Whitney test (two-tailed): $U = 98$, $p = 0.535$. $n = 12$ for the control, and $n = 19$ for the pubertal stress group. **b** Estradiol concentration at the age of P60. Two-tailed Mann–Whitney test: $U = 81$, $p = 0.4379$. Control: $n = 11$; pubertal stress: $n = 18$. **c** Progesterone concentration at the age of P60 in control subjects as well as females that were exposed to pubertal stress. Mann–Whitney test: $U = 82$, $p = 0.724$. $n = 10$ and $n = 18$ for control and pubertal stress, respectively. **d** Schematic representation of blood sampling procedures to measure the levels of corticosterone and DHEA by ELISA. EPM: elevated plus maze. Figure created with BioRender.com. **e** Levels of plasma corticosterone in control and pubertally stressed females under home cage conditions ($n = 17$ for the control group, and $n = 6$ for pubertal stress) and following exposure to the elevated plus maze ($n = 18$ for the control group, and $n = 12$ for the pubertally stressed females). Two-way RM ANOVA; Group effect: $F(1,28) = 2.247$,

$p = 0.145$; Treatment effect: $F(1,31) = 139.4$, $p < 0.0001$); Group × Treatment effect: $F(1,21) = 0.7097$). **f** Concentration of DHEA in plasma at the baseline (home cage; $n = 9$ for the control group, and $n = 5$ for the pubertal stress group) versus response (exposure to the EPM; $n = 15$ and $n = 11$ for the control and pubertal stress groups, respectively) of the HPA axis. Two-way RM ANOVA; group effect: $F(1,36) = 6.062$, $p = 0.0187$; treatment effect: $F(1,36) = 22.84$, $p < 0.0001$; Group × treatment effect: $F(1,36) = 0.0598$. **g** Corticosterone/DHEA ratio at the baseline ($n = 9$ and 5 for the control and pubertal stress group, respectively) and following exposure to the elevated plus maze (response: $n = 15$ and 11 for the control and pubertal stress group, respectively). Two-way RM ANOVA; group effect: $F(1, 25) = 1.674$, $p = 0.207$; Treatment effect: $F(1, 11) = 88.03$, $p < 0.0001$; group × treatment effect: $F(1,11) = 1.134$, $p = 0.309$. Data are presented as mean ± SEM. Source data are provided as a Source Data file.

comparisons test; male urine vs. water: $p = 0.0061$, female urine vs. water: $p = 0.871$, male urine vs. female urine: $p = 0.0412$) (Fig. 6q, Supplementary video 1).

To examine whether the activation of nNOS neurons by urinary olfactory cues is influenced by circulating sex steroid hormones, nNOS neurons were recorded under different hormonal conditions in the

same females (Fig. 6 and Supplementary Fig. s6). The activation of the nNOS neurons in response to male urine was the highest when female mice were treated with both estradiol and progesterone following ovariectomy (OVX + E2 + P4) (Fig. 6r and Supplementary Fig. s6q). In the presence of estradiol (OVX + E2), nNOS activation in male urine was significantly lower (Fig. 6i, j) in comparison to OVX + E2 + P4 condition

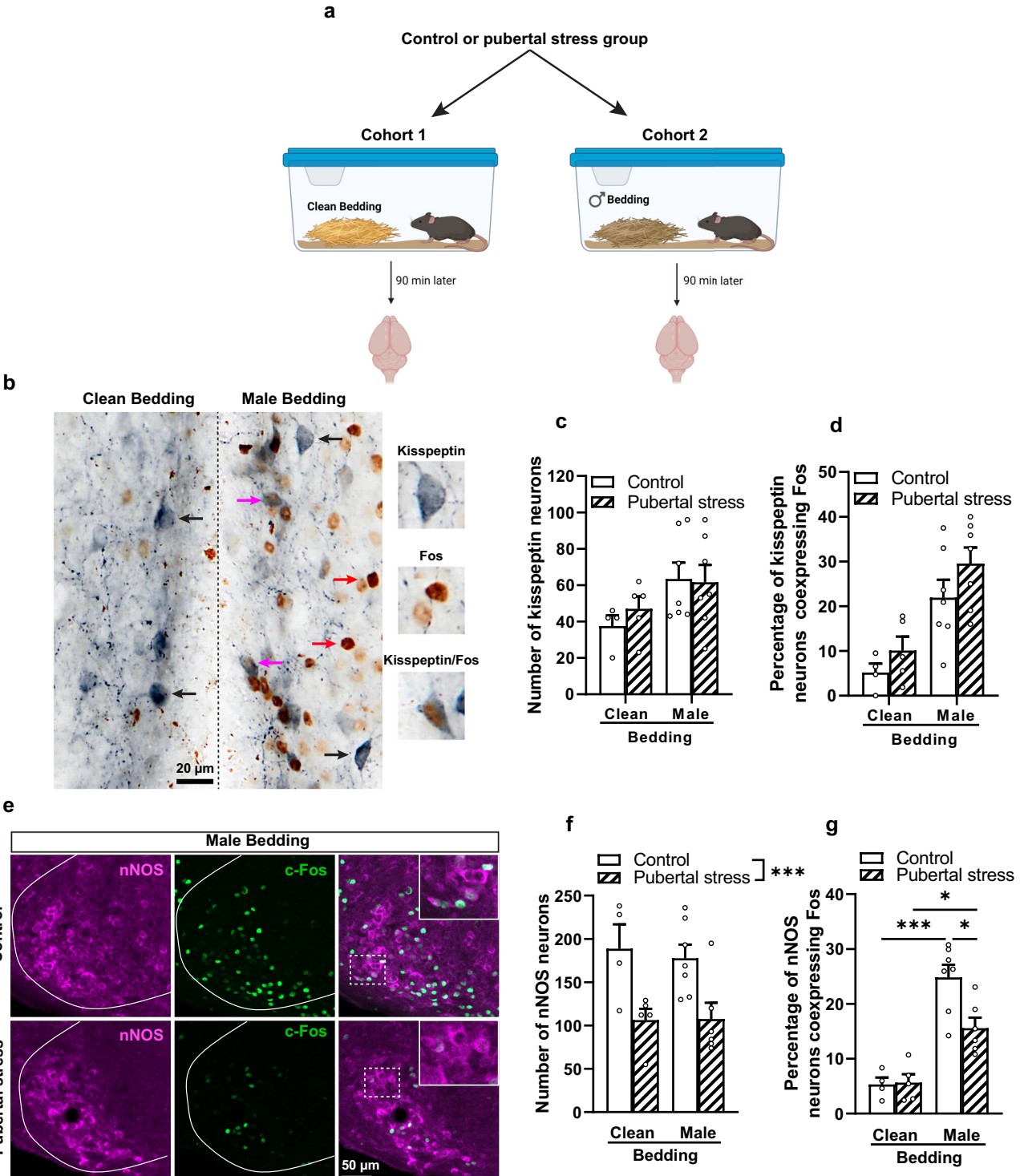

(Fig. 6a, b, r). A similar observation was made for females treated with progesterone alone (OVX + P4) (Supplementary Fig. s6a–d, q). No difference was observed in the activation of nNOS neurons in response to female urine in the OVX + P4 compared to the OVX + E2 + P4 condition (Supplementary Fig. s6i–l, r). Interestingly, removal of the estradiol implant followed by a two-week washout period did not have any effect on the activation of the nNOS neurons to olfactory cues (Fig. 6m, n). Indeed, no statistical difference (Bonferroni's multiple comparisons test following RM one-way ANOVA; $p > 0.999$) was observed in the peak activation of the nNOS neurons in response to male urine in ovariectomized females (OVX) compared to estradiol treatment (OVX + E2) (Fig. 6r). Regarding the response to female urine,

the activation of the nNOS neurons seemed to be independent of sex steroid hormones (Fig. 6c, d, k, l, o, p and Supplementary Fig. s6c, d). This was confirmed by statistical analysis revealing no differences in the activity of nNOS neurons to female urine under different hormonal conditions (RM one-way ANOVA; $F_{(1.129, 5.644)} = 1.369$; $p = 0.2968$) (Fig. 6s).

It should be noted that we noticed that the strength and the duration of the nNOS activation in response to male urine was smaller in sexually naïve females (Supplementary Fig. s6a, b) compared to sexually experienced female mice (Fig. 6a, b).

Using a similar approach as described above, we used stereotaxic surgery to deliver the GCaMP6s-expressing AAV in the VMHvl of

**Fig. 3 | Exposure to pubertal stress did not affect RP3V kisspeptin neurons and their activation by male odors whereas it reduced the number of the nNOS neurons in the VMHvl and their activation by male odors. a** Schematic representation of the protocol used to induce Fos expression in the neuronal circuit involved in sexual behavior. Figure created with BioRender.com. **b** Representative images of double labeling for kisspeptin and Fos in the RP3V following exposure to clean versus male bedding. Black arrows show examples of kisspeptin neurons, red arrows show examples of Fos expressing neurons, while pink arrows point towards examples of neurons coexpressing kisspeptin and Fos. **c** Number of kisspeptin neurons in the control ($n = 4$) and pubertal stress group ($n = 5$) groups following exposure to clean bedding, and in the control ($n = 7$) and pubertal stress ($n = 7$) after exposure to male bedding. Two-way ANOVA; Bedding effect: $F_{(1, 16)} = 4.968$, $p = 0.0381$; group effect: $F_{(1,19)} = 0.177$, $p = 0.679$; bedding × group effect: $F_{(1,19)} = 0.39$, $p = 0.539$. **d** Percentage of neurons coexpressing kisspeptin and Fos in the control ($n = 4$) and pubertal stress ($n = 5$) groups following exposure to clean bedding, and in the control ($n = 7$) and pubertal stress ($n = 7$) after exposure to male

bedding. Two-way ANOVA; Bedding effect: $F_{(1,19)} = 23.15$, $p = 0.0001$; group effect: $F_{(1,19)} = 2.7778$, $p = 0.1119$; interaction: $F_{(1,19)} = 0.13$, $p = 0.722$. **e** Representative confocal images showing nNOS/Fos immunofluorescence staining in the VMHvl. White dashed boxes indicate the regions of high magnification (**f**) Number of nNOS-immunoreactive neurons in the VMHvl of the control and pubertal stress groups following exposure to clean or male bedding. Two-way ANOVA: bedding effect: $F_{(1,18)} = 0.068$, $p = 0.796$; group effect: $F_{(1,18)} = 16.28$, $p = 0.0008$; bedding × group: $F_{(1, 18)} = 0.1088$, $p = 0.745$. ***$p < 0.001$. **g** Percentage of nNOS neurons expressing Fos following exposure to male bedding. Two-way ANOVA followed by Bonferroni multiple comparison test; Bedding effect: $F_{(1,18)} = 51.23$, $p < 0.0001$; Group effect: $F_{(1,18)} = 4.684$, $p = 0.044$; interaction: $F_{(1,18)} = 5.419$, $p = 0.0318$. *$p < 0.05$; ***$p < 0.001$. $n = 4$ for control group exposed to clean bedding, $n = 5$ for pubertal stress group exposed to clean bedding, $n = 7$ for control group exposed to male bedding, $n = 6$ for pubertal stress group exposed to male bedding. All data are presented as mean ± SEM. Corrections were made whenever multiple comparisons test is used. Source data are provided as a Source Data file.

nNOS::Cre females that were previously exposed to pubertal stress. (Fig. 7). We found that nNOS neurons activation by male urine in pubertally stressed females, but which remained sexually receptive (HR females) (Fig. 7c, d) was similar to the activation observed in control, non-stressed females (post-hoc test following one-way ANOVA; $p > 0.99$) (Fig. 7a, b, h). By contrast, MR females displayed a blunted activation of the nNOS neurons when exposed to male urine compared to the control ($p = 0.0116$) and the HR females ($p = 0.0513$) (Fig. 7e, f, h). No differences were observed between all groups in their nNOS neuronal responses to female urine ($F_{(2,15)} = 2.33$; $p = 0.131$) (Fig. 7i–p).

Taken together, these data revealed that the nNOS neurons located in the VMHvl were selectively activated by olfactory cues that are sexually relevant in female mice, and that their activation in response to male urinary odors was modulated by the synergistic effect of estradiol and progesterone. In addition, exposure to stress over puberty in female mice specifically blunted the activity of these neurons in response to male urinary odors.

Giving that pubertal stress impacted the response of nNOS neurons to ovarian hormones and that previous studies have identified the expression of estradiol receptor alpha (Erα) and progesterone receptor (PR) in the VMHvl, we checked whether pubertal stress alters the expression of Erα and PR in VMHvl nNOS neurons. Immunohistochemical analysis revealed that 60% of nNOS neurons in the VMHvl are expressing PR (Fig. 8a, b). In addition, no difference in the number of PR-expressing neurons was observed in HR and MR females compared to the control group (Fig. 8d). However, we found a significant reduction in the number of nNOS neurons colocalizing with PR in MR females compared to the control group (Fig. 8e).

Analysis of nNOS/ERα coexpression revealed that 85% of nNOS neurons are expressing Erα in the VMHvl (Fig. 8f, g), and that pubertal stress had no effect on the number of neurons expressing ERα in the same nucleus (Fig. 8i). Further analysis indicated a significant reduction in the number of nNOS neurons colocalizing with ERα in MR females compared to the control subjects (Fig. 8j). Taken together, these data indicate that the dysregulation in the VMHvl is specific to the expression of nNOS enzyme without impacting the expression of Erα or PR in the same neurons.

**Supplementation with a NO donor rescues sexual behavior in females exposed to pubertal stress**

Next, we determined whether peripheral administration of kisspeptin (Kp-10) or SNAP, a nitric oxide donor (NO), in female mice exposed to pubertal stress might rescue their disrupted lordosis behavior. We found that subcutaneous administration of Kp-10 or SNAP had no effect on lordosis behavior in primed (E2 + P4) HR females, probably due to a ceiling effect (Fig. 9b). Similarly, administration of Kp-10 to MR females had no significant effect on their levels of lordosis

behavior (Fig. 9b). Interestingly, peripheral injection of SNAP in MR females significantly stimulated lordosis behavior compared to vehicle ($p = 0.003$; Bonferroni's multiple comparisons test following RM one-way ANOVA) (Fig. 9b).

To assess if central infusion of SNAP into the VMHvl of female exposed to pubertal stress might recue their sexual behavior, females were implanted with a bilateral guide cannula aimed at the VMHvl (Fig. 9c). At the day of drug testing, females received a subcutaneous injection of progesterone. Two hours later, subjects received either SNAP or vehicle infusion and were then assessed for lordosis behavior 1 h later (Fig. 9a). Infusion of SNAP into the VMHvl of HR females tended to increase their lordosis behavior but the effect was not significant ($p = 0.2721$; Bonferroni's multiple comparisons test following RM two-way ANOVA) (Fig. 9d). By contrast, administration of SNAP significantly increased the lordosis quotient in MR females ($p = 0.0362$; Bonferroni's multiple comparisons test) (Fig. 9d), albeit not the level observed in pubertally stressed HR females, suggesting a partial rescue of lordosis behavior in these females.

To understand how SNAP might stimulate lordosis behavior in MR-receptive females, we measured the number of phosphorylated nNOS neurons using pnNOS immunostaining. Following peripheral SNAP or vehicle administration, control and MR females were perfused 15 min later. Immunostaining for pnNOS showed a significant increase in the number of pnNOS immunoreactive neurons following SNAP treatment in both the VMHvl (Fig. 9e, g) and the PVN (Fig. 9f, g) nuclei, with no significant difference between control subjects and MR females, indicating an activation of nNOS neurons. Considering that MR females are characterized by a reduced number of nNOS neurons, these results suggest that NO supplementation can modulate the activity and the expression of nNOS. This further demonstrates that pubertal stress did not induce a loss of nNOS neurons but rather a reduction in the expression of the nNOS enzyme in these neurons.

## Discussion

Puberty is a critical developmental stage during which a cascade of biological events leads to the maturation of female reproductive behavior. The development of the brain during this period makes some neural circuits particularly sensitive to stress leading to a disruption of sexual behavior. We found that exposure to stress over puberty induced a long-lasting alteration of the female reproductive cycle and sexual performance. Mice exposed to pubertal stress showed disrupted lordosis behavior, but sexual preferences were not affected. These observations were not associated with an increased state of anxiety or depression-like behaviors. Further investigations revealed that the effect of chronic pubertal stress on female sexual performance was associated with a decreased number and activation by sexual cues of nNOS neurons located in the VMHvl.

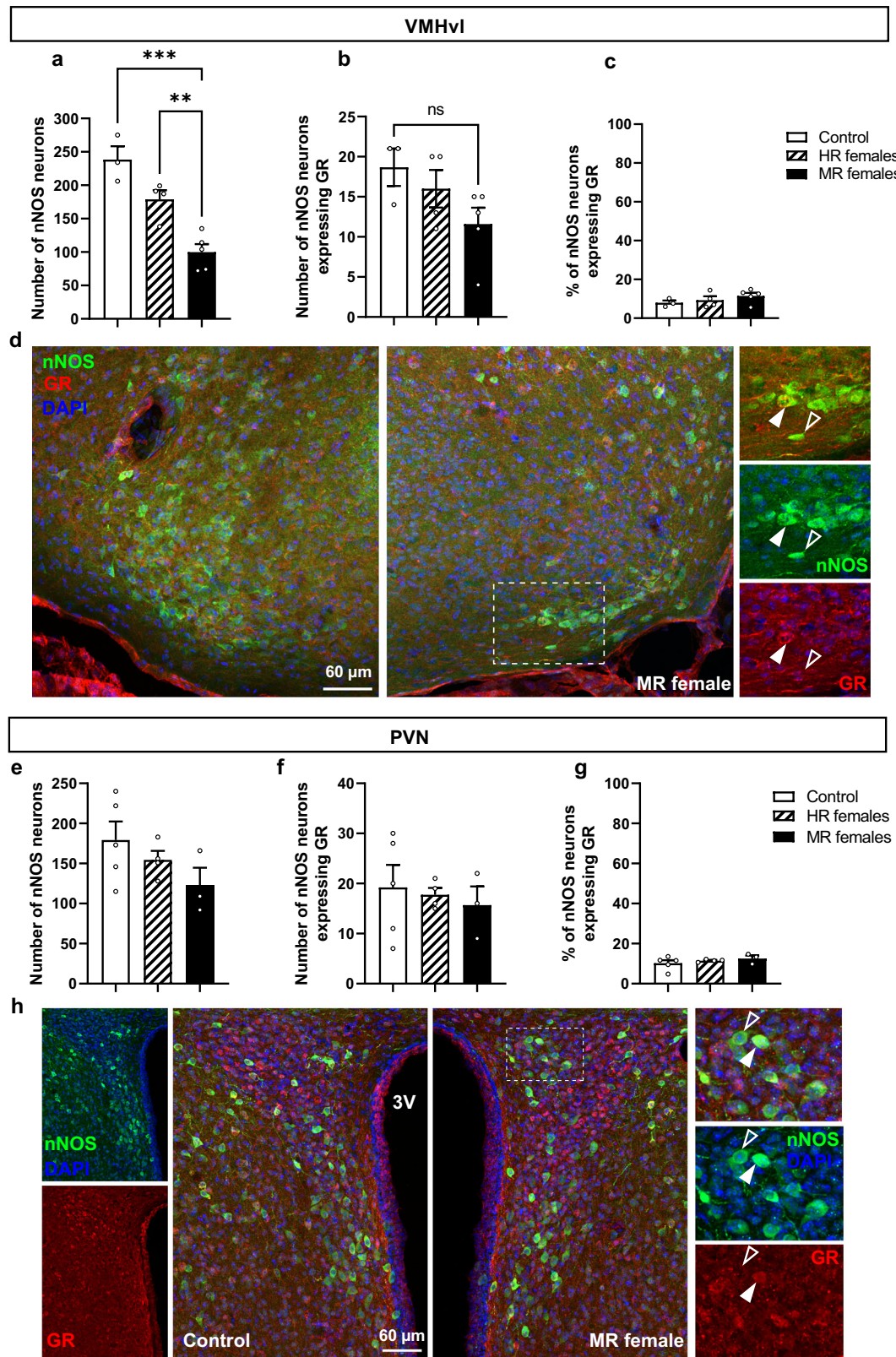

Lordosis behavior was markedly disrupted in a subpopulation of females exposed to pubertal stress; this concerned around 60% of the subjects. Similar proportions of HR and MR females were observed in multiple experiments in the present study, confirming the consistency and the reproducibility of this effect of pubertal stress on female sexual behavior. Our results are in line with a previous study[16] showing that social isolation during puberty affects lordosis behavior. The

uneven manifestation of an effect of stress in MR females compared to HR females can probably be associated with the intensity of the used stressor. Indeed, previous studies (reviewed in ref. 5) have shown that not all types of stressors can induce similar effects on female sexual behavior. Multiple stressors were actually tested, including restrain stress, heat, and light stress, food deprivation, immune challenge with the bacterial endotoxin LPS, as well as social isolation. Although these

**Fig. 4 | Pubertal stress had no effect on the subpopulation of nNOS neurons in the VMHvl and PVN coexpressing glucocorticoid receptor (GR). a** Number of nNOS neurons in the VMHvl. One-way ANOVA; $F_{(2,9)} = 22.47$, $p = 0.00031$. **$**p = 0.0086$; $***p = 0.0003$. b** the number of nNOS expressing GR. One-way ANOVA: $F_{(2,9)} = 2.519$, $p = 0.1353$. ns = not significant. **c** Percentage of nNOS expressing GR in the VMHvl. One-way ANOVA: $F_{(2,9)} = 1.06$, $p = 0.385$. Control ($n = 3$), highly receptive (HR) females ($n = 4$), Minimally receptive (MR) females ($n = 5$). **d** Representative confocal images showing nNOS/GR immunostaining in the VMHvl in a control subject vs a MR female. White dashed box indicates the region of high magnification. Full arrow shows an example of colocalization between nNOS and GR, whereas hollow arrow points to nNOS expression only. **e** Number of nNOS neurons in the PVN. One-way ANOVA: $F_{(2,9)} = 1.789$, $p = 0.221$. **f** Number and (**g**) percentage of nNOS neurons in the PVN expressing GR. One-way ANOVA; number of nNOS neurons expressing GR ($F_{(2,9)} = 0.203$, $p = 0.819$), Percentage of nNOS expressing GR ($F_{(2,9)} = 0.826$, $p = 0.468$). Control ($n = 5$), HR females ($n = 4$), MR females ($n = 3$). **h** Representative confocal images showing nNOS/GR immunostaining in the PVN in a control vs a MR female. White dashed box indicates the region of high magnification. Full arrow shows an example of colocalization between nNOS and GR, whereas hollow arrow points to nNOS expression only. All data are presented as mean ± SEM. Corrections were made whenever multiple comparisons test is used. Source data are provided as a Source Data file.

stressors induced quite high blood corticosterone levels during the period of exposure, only social isolation and immune challenge resulted in a decreased response to estradiol and progesterone in adulthood.

As in the present study, previous work has shown that socialization after puberty did not rescue female sexual behavior[16], suggesting permanent effects of social isolation during puberty on later sexual behavior. This impairment was found in ovariectomized females that were supplemented with estradiol and progesterone, suggesting that the observed defect is not associated with abnormal levels of sex steroid hormones and may thus result from developmental disruptions.

In the present study, a significant correlation was found between sexual receptivity and baseline CORT/DHEA ratio. This observation is in line with a previous clinical study by Basson and colleagues[8] showing that women with low sexual desire have abnormal levels of stress hormones indicating a disrupted functioning of the HPA-axis. However, it is not clear whether the disrupted HPA axis affects sexual behavior, or whether it simply reflects previous exposure to chronic stress. In the present study, since the evaluation of estradiol concentrations was performed during pubertal development only at the age of P40, we cannot exclude a transient effect of stress on the levels of sex steroid hormones. However, it seems unlikely that this was the case since the number of kisspeptin neurons in the RP3V was not affected and it has been shown that this neuronal population is particularly sensitive to estradiol during pubertal development[22,23], suggesting that stress exposure at the age of puberty had no effect on estradiol levels. Furthermore, our data showed that females exposed to pubertal stress spent more time in metestrus and diestrus compared to the other stages of the estrous cycle, a phenotype that was recently observed in brain-specific ERα knockout mice[34] as well as in nNOS signaling deficient mice[35].

One of the major findings here is that exposure to pubertal stress by social isolation reduced the number of neuron-expressing nNOS in the VMHvl. The mechanism by which stress disrupted this neuronal population is unknown. According to previous reports[36], the projections of the VMH are visible as early as embryonic day 10.5, suggesting that the formation of the VMH circuit starts before birth. In addition, no difference was found in the number of VMHvl nNOS neurons between adult subjects and pups at postnatal day 11[37], indicating that pubertal stress could not have interfered with their development. In addition, we found that SNAP treatment resulted in a similar number of activated nNOS cells as measured by pnNOS staining in both control and MR females. Therefore, it could be speculated that the observed effect is probably associated with disrupted gene transcription. Indeed, it has been reported that stress might alter gene transcription leading to aberrant neuronal functioning[38–40], which might explain the reduced nNOS-expressing neurons in the present study. Alternatively, but less probable, is an induction of nNOS apoptosis by glucocorticoids, as we observed that around 10% of nNOS neurons express GR. It has been reported that stress can influence neuronal survival in the hippocampus through excess activation by glucocorticoids receptors which induced apoptosis[41–43]. However, our study indicated that there is no reduction in the number of nNOS neurons coexpressing GR receptor, suggesting that there is no involvement of GR signaling in the effect of pubertal stress on the VMHvl nNOS.

The fact that not all VMHvl nNOS neurons in the present study are impacted by the stress exposure suggests the existence of multiple subpopulations with different characteristics and distinct involvement in reproductive behavior. Indeed, it has been shown that the female VMHvl contains two anatomically distinguishable subdivisions that showed differential gene expression, projections, and activations following mating and aggression[44]. It has been previously reported that ERα-expressing neurons[44] and PR-expressing neurons[45] in the VMHvl are involved in female sexual behavior. Even though we found that 60% and 85% of the VMHvl nNOS neurons are coexpression PR and Erα, respectively, there is a specific reduction in the number of nNOS neurons without impacting the expression of either PR or ERα, suggesting a reduction in the nNOS enzyme production in these neurons.

Our calcium recordings showed that the nNOS neurons located in the VMHvl were specifically activated by male olfactory cues. Longitudinal within-subject recordings revealed that the activation of the nNOS neurons was increased by progesterone in response to male urine. Previous studies have shown that male olfactory cues are necessary for the expression of lordosis behavior in female mice. For instance, ablation of the vomeronasal organ strongly disrupted lordosis behavior in ovariectomized females that were primed with estradiol and progesterone[19,46]. Therefore, the reduced activation of the nNOS neurons in response to male olfactory cues following pubertal stress may indicate a disrupted integration of olfactory inputs arriving at the VMHvl, subsequently leading to lower sexual performance.

The observation of an increased activity of VMHvl nNOS neurons in females treated with both estradiol and progesterone but not with either one of them alone raises the question about the mechanisms by which the nNOS neurons become more responsive to male-derived olfactory cues in the presence of both ovarian hormones. It can be hypothesized that the activation of these neurons can probably be associated with the onset of an event upstream of VMHvl nNOS neurons following synergic effects of E2 and P4, which facilitate or increase the excitatory inputs received from male-responsive-cells such as the RP3V kisspeptin neurons. Interestingly, removal of the estradiol implant followed by two weeks of a washout period did not alter the excitability of the nNOS neurons. This observation might be associated with sexual experience. Previous data have shown that sexually naïve, hormone-primed, female mice are not receptive to mating attempts, while repeated sexual experience with males and concurrent treatment with estradiol and progesterone gradually increases sexual receptivity over the course of multiple trials[47]. This suggests that experience can adjust and modify the neuronal circuits regulating sexual behavior. In addition, recent evidence indicates that VMHvl responses to the investigation of male and female subjects overlap initially in sexually naïve males, whereas a brief sexual experience with females induced a considerable divergence in the VMHvl responses towards male and female subjects[48]. Interestingly, we observed a difference in the activation of VMHvl nNOS neurons to male olfactory cues, in sexually experienced subjects compared to sexually naive

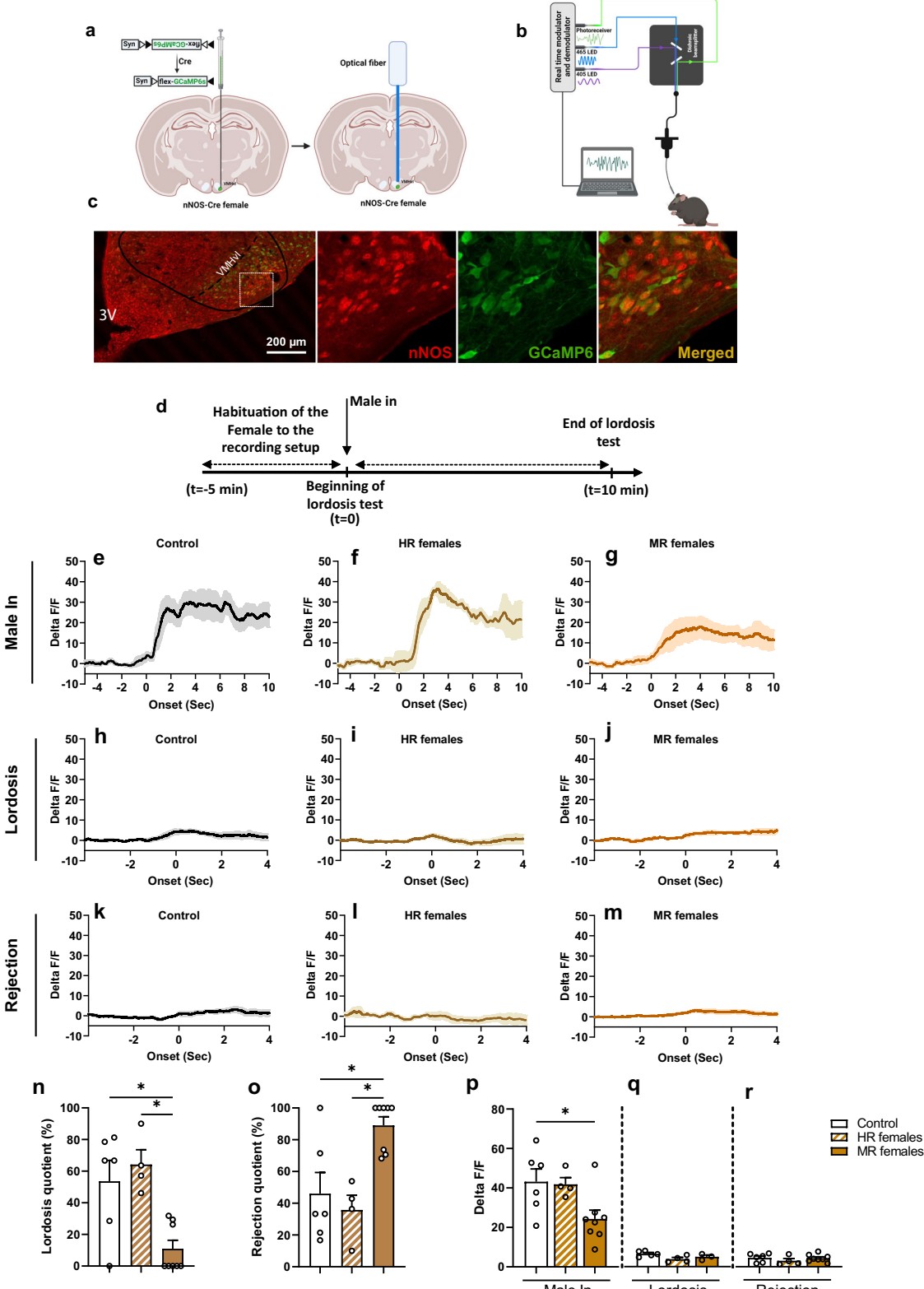

females. The activation was strong and lasted several seconds in sexually experienced females, while nNOS response was small and quickly disappeared in sexually naive females, suggesting that sexual experience can modulate the activity of the VMHvl nNOS neurons in response to male olfactory cues. Intriguingly, no significant variation in VMHvl nNOS neuronal activity was observed when female mice expressed lordosis or rejection behavior. Multiple studies, including VMHvl lesions[49], drug deliveries[20], hormone supplementations[50], and

measures of Fos expression following mating[19] have suggested a direct involvement of the VMHvl in the expression of lordosis behavior. However, it is not clear whether the observed effects were associated with the expression of lordosis itself or an enhancement/disruption of olfactory processing. For instance, it cannot be concluded from these studies whether Fos expression in the VMHvl following mating is associated with the expression of lordosis or an olfactory stimulation due to the close proximity of the male. It might thus be possible that

**Fig. 5 | VMHvl nNOS neurons are particularly activated during interaction with the stimulus male. a** Schematic representation of stereotaxic surgery to inject the AAV virus for expressing the GCaMP6s calcium sensor and fiber optic implantation into the VMHvl. Figure created with BioRender.com. **b** Schematic representation of the fiber photometry setup used to record the VMHvl nNOS neurons. Figure created with BioRender.com. **c** Representative images of Cre-dependent GCaMP6s expression in nNOS neurons. **d** Timeline representation of lordosis behavior test to measure the activity of VMHvl nNOS neurons. All females were submitted to 2 lordosis pre-tests before undergoing the fiber photometry recording during the third session. **e–g** Peri-event time histograms and (**p**) peak delta F/F of the nNOS neurons activation when the stimulus male was introduced for lordosis behavior in control group ($n = 6$), highly receptive (HR) ($n = 4$) and minimally receptive (MR) ($n = 8$) females. One-way ANOVA followed by Bonferroni: $F(2,15) = 0.0293$.

$*p = 0.0485$; Bonferroni's post hoc test. **n** Percentage of Lordosis quotient (Kruskal–Wallis test followed by Dunn's multiple comparison test: H(3) = 8.797, $p = 0.0059$; $*p = 0.0481$ and 0.0359 for control vs MR females and HR females vs MR females, respectively), (**o**) and Percentage of rejection during lordosis test (Kruskal–Wallis test: H(3) = 8.797, $p = 0.0059$) in control ($n = 6$), HR ($n = 4$) and MR females ($n = 8$) at the day of fiber photometry recording; $*p < 0.05$; Dunn's multiple comparisons test. **h, i, j, q** Activation of the nNOS neurons when females from control group ($n = 5$), HR females ($n = 4$) and MR females ($n = 3$) performed lordosis behavior; only subjects that expressed lordosis were included in the analysis. **k, l, m, r** Activation of the nNOS neurons when control ($n = 6$), HR ($n = 4$) and MR ($n = 8$) females rejected the mount attempts by the stimulus male. Data are presented as mean ± SEM. Corrections were made whenever multiple comparisons test is used. Source data are provided as a Source Data file.

nNOS neurons in the VMHvl are a necessary relay for processing sexual olfactory cues that will lead to the expression of lordosis behavior through other downstream neuronal targets, for example the periaqueductal gray.

In summary, our findings demonstrate the importance of nNOS neurons located in the VMHvl as an important node in the processing of olfactory cues necessary for the expression of female sexual behavior. In addition, we confirm that puberty is a critical period for the development and organization of female sexual behavior. Adverse events during this particular period might have long-lasting effects on sexual performance, which are caused by disrupted integration of male olfactory cues by the VMHvl nNOS neurons.

## Methods
### Subjects
All experiments were performed using Wild-type *C57BL/6J* mice obtained from Janvier Labs, or nNOS::Cre (*C57BL/6J* background) purchased from the Jackson Laboratory (JAX stock #017526) and bred in our animal facility. Animals were kept under standard laboratory conditions until crossed to generate the experimental subjects. Experimental and stimulus animals were housed under conditions of controlled temperature ($22 \pm 2\,°C$) and lighting (12-h light, 12-h dark cycle; lights off at 08:00 h and lights on at 20:00 h) with food and water available ad libitum. Stimulus males were housed individually, whereas stimulus females were kept group housed in 3 to 4 subjects per cage. All behavioral tests were conducted in the dark phase of the light cycle. Experiments were approved by the ethics committee of the University of Liège (dossier number #20-2258).

### Stress procedure
Here we took advantage of the importance of social interaction for rodents and used social isolation as a stressor which has been shown in multiple studies to induce strong stress-related responses[16,18,51–53]. To induce pubertal stress, mice were exposed to chronic social isolation by housing them individually from the age of weaning, i.e., postnatal day 21 (P21) until P50. At the end of stress exposure, females were housed back together in groups of 3 to 4 subjects per cage. Control subjects were weaned at the age of P21, and group housed in 3 to 4 subjects per cage. This procedure was slightly adapted from[16] showing that social isolation resulted in decreased hormonally induced female sexual behavior, by limiting the social isolation to the pubertal period only.

### Vaginal opening and reproductive cycle
Weaned female mice were checked daily for vaginal opening. Vaginal cytology was assessed daily in early adulthood (P60) by dipping a ring-shaped stainless-steel spatula in saline, which was then gently used for swabbing the vaginal canal. Samples were transferred to a microscope slide, air dried and stained using 0.1% thionine blue. Estrous cycle stage was assessed using previously established criteria[54].

### Blood sampling and ELISA assays
To evaluate the levels of corticosterone and DHEA, tail blood was collected under two conditions: home cage as baseline and 30 min following exposure to the elevated plus maze to evaluate HPA axis response. Blood samples were collected only when the females were in the proestrous or estrous stage of the estrous cycle. To avoid the circadian peak and the diurnal variation in corticosterone, samples were collected between 14:30 h and 16:00 h in heparin-coated capillaries (Roth, TX80.1), and then centrifuged at $3000\,g$ for 15 min. The plasma was collected and stored at $-80\,°C$ until the assay was performed. Samples were diluted in assay buffer and analyzed according to the manufacturer's instructions (Enzo life sciences; corticosterone Elisa kit: ADI-900-097; DHEA Elisa kit: ADI-900-093).

To measure progesterone and 17β-estradiol concentrations, tail blood was collected in heparin-coated capillaries as well. Hormones concentration was determined by ELISA (Demeditec Diagnostics; estradiol Elisa kit: DE2693; Progesterone Elisa kit: DE1561) according to the manufacturer's instructions. The absorbance was read at 405 nm using the MultisKan Ascent microplate reader and the values were calculated based on the standard curve.

### Ovariectomy and hormone supplementation
Unless otherwise stated, females were ovariectomized in adulthood. Anesthesia was induced by an intraperitoneal injection of a mixture of ketamine (Nimatek, 80 mg/kg) and medetomidine (Domitor, Pfizer, 1 mg/kg). Then, both ovaries were removed, and an estradiol implant was placed subcutaneously in the back. The implant consisted of a 5 mm long capsule (outer diameter 2.41 mm, inner diameter 1.57 mm) filled with a 1:1 mixture of 17beta-estradiol (ICN biomedicals Inc., 101565) and cholesterol (Sigma, C8667). At the end of the surgery, all mice received a subcutaneous injection of atipamezole (Nacrostop, 4 mg/kg) to accelerate recovery by antagonizing medetomidine-induced effect. All animals were allowed to recover on a heating pad and were returned subsequently to their home cage[19,20]. In order to induce sexual receptivity at the day of testing, females received a subcutaneous injection of progesterone (500 µg, P0130, Sigma) 2 to 3 h before the test.

### Subcutaneous kisspeptin and SNAP treatments
To determine whether kisspeptin (Kp-10) or S-nitro-*N*-acetyl-DL-Penicillamine (SNAP) can rescue sexual behavior in pubertally stress females, a subcutaneous injection of Kp-10 (4243, Tocris) or SNAP (N3398, Sigma) was delivered 1.5 h before the behavioral test. Kp-10 was used at the dose of 0.13 mg/kg, while SNAP (8 mg/kg) was combined with BAY 41-2272 (10 mg/kg; B8810, Sigma) to increase the efficiency of the NO donor[19,20].

### Stereotaxic surgery for cannula implantation and drug infusion
Stereotaxic surgery for implanting a bilateral guide cannula aimed at the VMHvl was performed under isoflurane anesthesia. Mice were

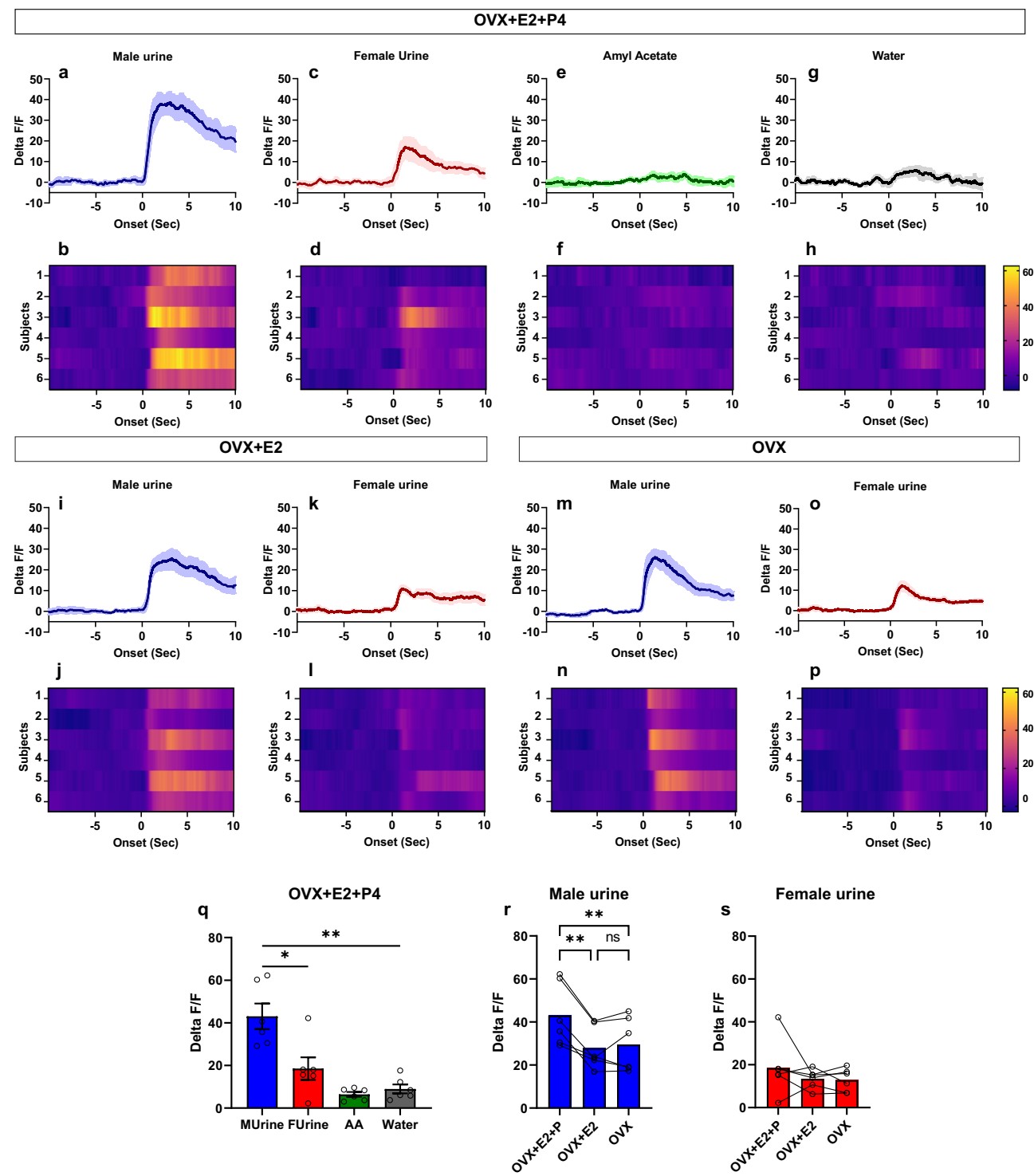

**Fig. 6 | nNOS neurons located in the VMHvl are activated by male olfactory cues and modulated by the synergic effect of estradiol and progesterone.** Peri-event time plots and heat maps of nNOS neurons to male urine (**a**, **b**), female urine (**c**, **d**), amyl acetate (**e**, **f**) or water (**g**, **h**) in ovariectomized females that were primed with estradiol and progesterone (OVX + E2 + P4). Peri-event time histogram of nNOS activation following exposure to male urine (**i**) or female urine (**k**) in ovariectomized females that were treated with an estradiol implant (OVX + E2). Heat map of nNOS activation during exposure to male urine (**j**) or female urine (**l**) in ovariectomized females with and estradiol implant (OVX + E2). Peri-event time histograms and heat maps of nNOS activation in response to male (**m**, **n**) or female (**o**, **p**) urine in ovariectomized subjects following removal of the estradiol implant and a washout period of two weeks. **q** Comparison of peak delta F/F of the VMHvl nNOS neurons in response to different olfactory cues. RM one-way ANOVA, followed my Bonferroni multiple comparisons test: $F(2.016, 10.08) = 24$, $p = 0.0001$. *$p = 0.0412$; **$p = 0.0061$. **r** Comparison of nNOS activation in response to male urine under different hormonal conditions. RM one-way ANOVA followed by Bonferroni's post hoc test: $F(1.838, 9.188) = 23.0$, $p = 0.0003$. **$p = 0.0082$ and $p = 0.0044$ for OVX + E2 + P4 vs OVX + E2 or OVX, respectively. **s** Activation of nNOS neurons in response to female urine under different hormonal conditions. RM one-way ANOVA: $F(1.129, 5.644) = 1.369$, $p = 0.296$. OVX (ovariectomized); E2 (estradiol); P4 (progesterone); AA (amyl acetate); MUrine (male urine); FUrine (female urine). All subjects ($n = 6$) used in this experiment were sexually experienced. Data are presented as mean ± SEM. Adjustments were made whenever multiple comparisons test is used. Source data are provided as a Source Data file.

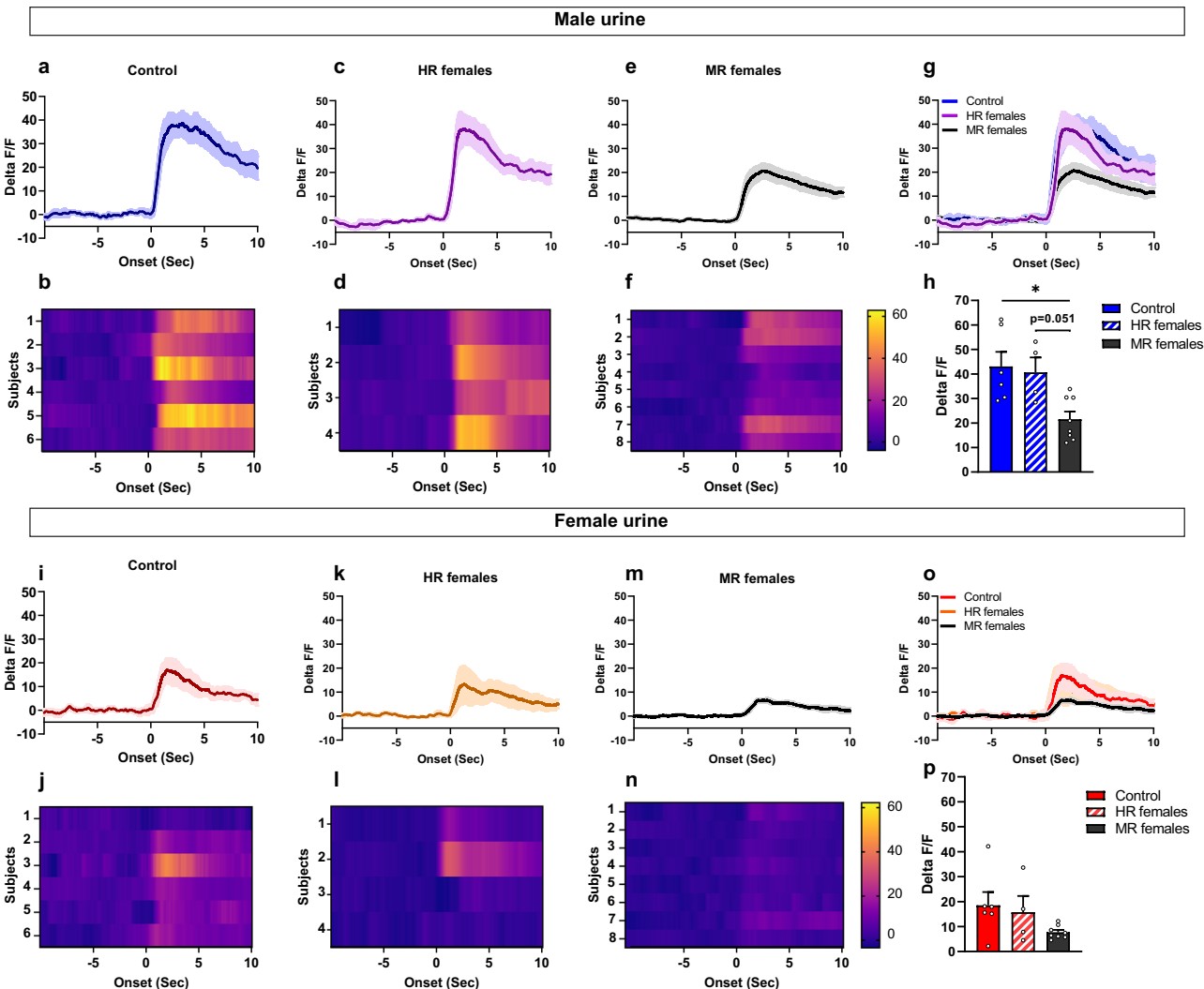

**Fig. 7 | Activation of the VMHvl nNOS neurons in response to male olfactory cues is specifically disrupted in MR females that were exposed to pubertal stress.** Peri-event time plots and heat maps of the VMHvl nNOS activation in response to male olfactory cues in the control group (**a**, **b**), pubertally stressed females that were sexually receptive (HR females) (**c**, **d**), and pubertally stressed female that were minimally sexually receptive (MR females) (**e**, **f**). **h**, **g** Comparison of peak delta F/F nNOS activation in all groups in response to male urine. One way ANOVA ($F(2,15) = 6.947$, $p = 0.0073$), followed by Bonferroni's post hoc test; *$p = 0.0116$. Peri-event time plots and heat maps of the nNOS neurons responses to female urine in control females (**i**, **j**), stressed but sexually receptive females (HR

females) (**k**, **l**), and stressed females minimally sexually receptive (MR females) (**m**, **n**). **o**, **p** Comparison of nNOS activation in response to female urine. One-way ANOVA $F(2,15) = 2.33$, $p = 0.131$. All subjects were ovariectomized, implanted with an estradiol implant and primed with progesterone 2 to 3 h before the fiber photometry recordings. Control ($n = 6$), HR females ($n = 4$), MR females ($n = 8$). The control group used here is similar to the one used in the previous experiment presented in Fig. 6. All groups were sexually experienced. Data are presented as mean ± SEM. Corrections were made whenever multiple comparisons test is used. Source data are provided as a Source Data file.

placed in a motorized stereotaxic frame (Neurostar, Stereodrive). Following skull exposure, two small holes were drilled to allow the insertion of a double guide cannula (PlasticsOne, C235GS-51/2/SPC, 4.75 mm) using the following stereotaxic coordinates: AP: -1.3 mm, ML: ±0.6 mm, DV: 4.75 mm. To avoid damaging the VMHvl, those coordinates were chosen to reach a depth of 1 mm above the target brain area. Subsequently, the guide cannula was secured in place using cyanoacrylate and dental acrylic cement (Unifast Trad, GC Inc.). Mice were allowed to recover for at least 10 days before any behavioral assessments. For behavioral testing, micro-infusions of 1 µl of vehicle or of a cocktail of SNAP (8 µg/kg; Sigma, N3398) and the guanylate cyclase agonist BAY-41-22711 (0.96 µg/kg; Sigma, B8819) were delivered using a 10 µl Hamilton syringe (65460-05) at a rate of 0.25 µl /min. The injection needle, which extended 1 mm past the end of the guide cannula, was left in place for 3 min to allow drug diffusion. Animals

were placed back in their home cage to recover for 60 min before undergoing the behavioral test.

**Viral injections and fiber optic implantation**
nNOS::Cre heterozygous females were anesthetized with isoflurane and placed into a stereotaxic frame. 200 nl of AAV1-Syn-Flex-GCaMP6s.WPRE.SV40 ($1 \times 10^{13}$ GC/mL) (a gift from Douglas Kim & GENIE project, Addgene viral prep #100845-AAV1) was injected unilaterally into the VMHvl using the following coordinates (AP: -1.3 mm; ML: +0.75 mm; DV: 5.75 mm) and at a speed of 50 nl/min. A barosilicate optical fiber with a zirconia ferrule (400 µm core, NA0.66, Doric MFC_400/430-0.66_5.5mm_ZF2.5(G)_FLT) was implanted 250 µm above the injection site and secured in place with dental cement (Unifast Trad, GC Inc.). All subjects were given 3 weeks for post-surgical recovery and viral expression before experiments were performed.

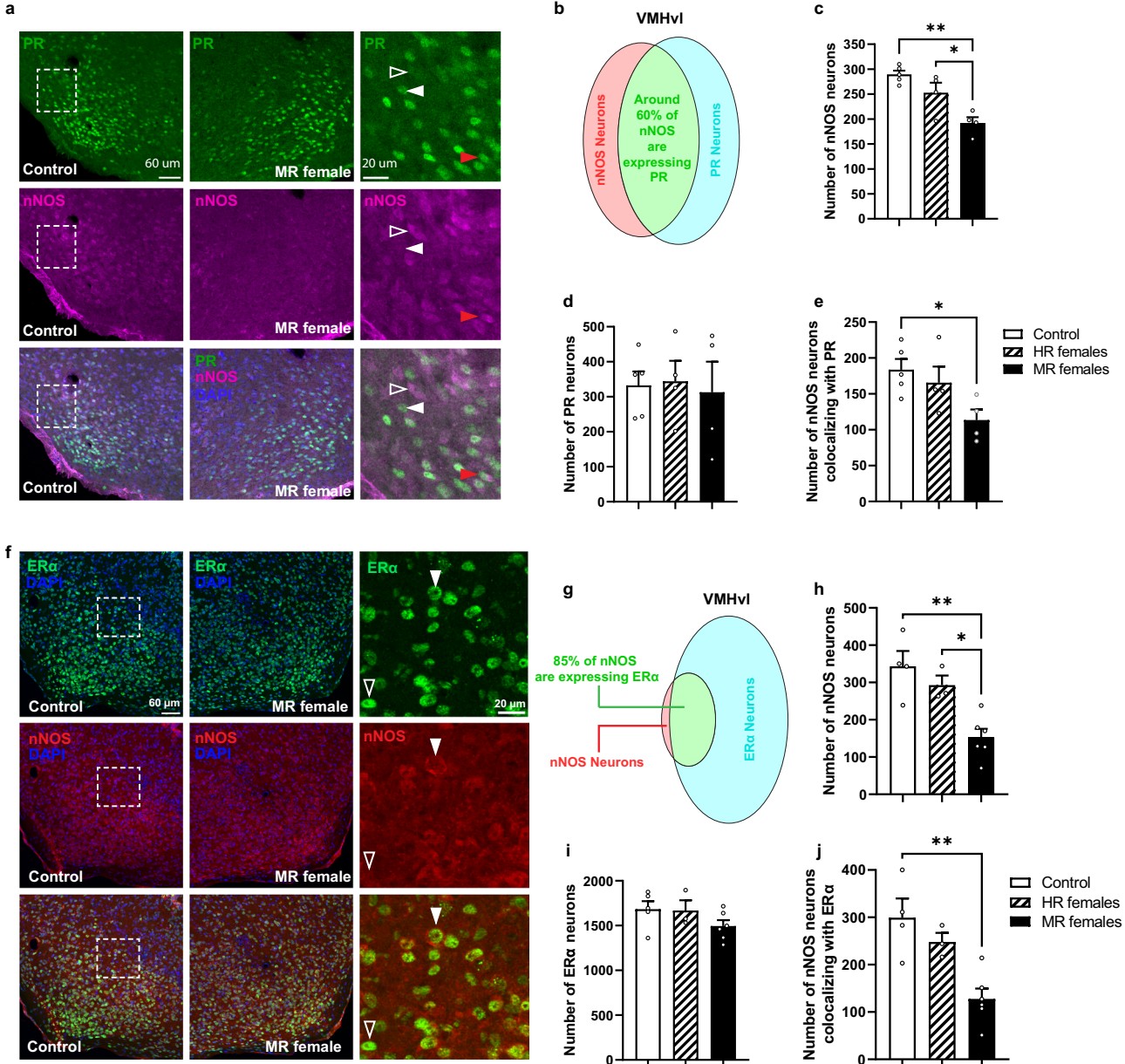

**Fig. 8 | Pubertal stress reduced the number of nNOS neurons in the VMHvl without affecting progesterone receptor (PR) or estradiol receptor alpha (ERα) populations. a** Representative confocal images showing nNOS and PR coexpression in the VMHvl in control and minimally receptive (MR) females. White dashed box shows the region of high magnification. Red arrow shows an example of a nNOS neuron expressing PR. White arrow indicates an example of a PR neuron that is not coexpressing nNOS, while white hollow arrow points to nNOS expression only. **b** Venn diagram representing the two populations of nNOS and PR neurons in the VMHvl. **c** Number of nNOS neurons in the VMHvl. One-way ANOVA; $F(2,10) = 131.91$, $p = 0.0013$. *$p = 0.0331$, **$p = 0.011$; Bonferroni's multiple comparisons test. **d** Number of PR neurons in the VMHvl. One-way ANOVA: $F(2,10) = 0.06$, $p = 0.941$. **e** number of nNOS neurons colocalizing with PR. One-way ANOVA: $F(2,10) = 4.329$, $p = 0.0442$. *$p = 0.0491$; Bonferroni's multiple comparisons test. $n = 5$ for control, $n = 4$ for HR females, and $n = 4$ for MR females. **f** Representative confocal images

showing nNOS and ERα coexpression in the VMHvl in control and MR females. White dashed box shows the region of high magnification. Solid white arrow shows an example of ERα expressing neuron coexpressing nNOS. White hollow arrow indicates an example of a nNOS neuron that is coexpressing ERα. **g** Venn diagram representing the distribution of nNOS and ERα and the percentage of their coexpression in the VMHvl. **h** Number of nNOS neurons in the VMHvl. One-way ANOVA followed by Bonferroni's post hoc test: $F(2,10) = 11.91$, $p = 0.0023$. *$p = 0.0331$, **$p = 0.0028$; Bonferroni's multiple comparisons test. **i** Number of ERα neurons in the VMHvl; $F(2,11) = 1.698$, $p = 0.227$. **j** Number of nNOS neurons colocalizing with ERα in the VMHvl. $F(2,10) = 10.52$, $p = 0.0035$. **$p = 0.004$; Bonferroni's post hoc test. Control ($n = 4$), HR females ($n = 3$), MR females ($n = 6$). All data are presented as mean ± SEM. Adjustments were made whenever multiple comparisons test is used. Source data are provided as a Source Data file.

## Fiber photometry

The GCaMP6s calcium sensor was imaged using a dual-wavelength fiber photometry system (Tucker-Davis Technologies, RZ10x). Two built-in LUX LED drivers (Lx405 and Lx465) were used to produce excitation wavelengths at 405 nm and 465 nm. To avoid cross talk between the two channels, each one was modulated at a different

frequency (210 Hz for 405 nm and 330 Hz for the 465 nm). Excitation lights were combined and filtered by a fluorescence minicube (Doric, FMC4, IE(400-410)_E(460-490)_F(500-550)_S). The combined excitation lights were delivered through a 400 nm core, 0.57 NA, low-autofluorescence monofiberoptic patchcord (Doric, MFP_400/430/1100-0.57_FCM-MF2.5_LAF) to the subject's fiber implant. The

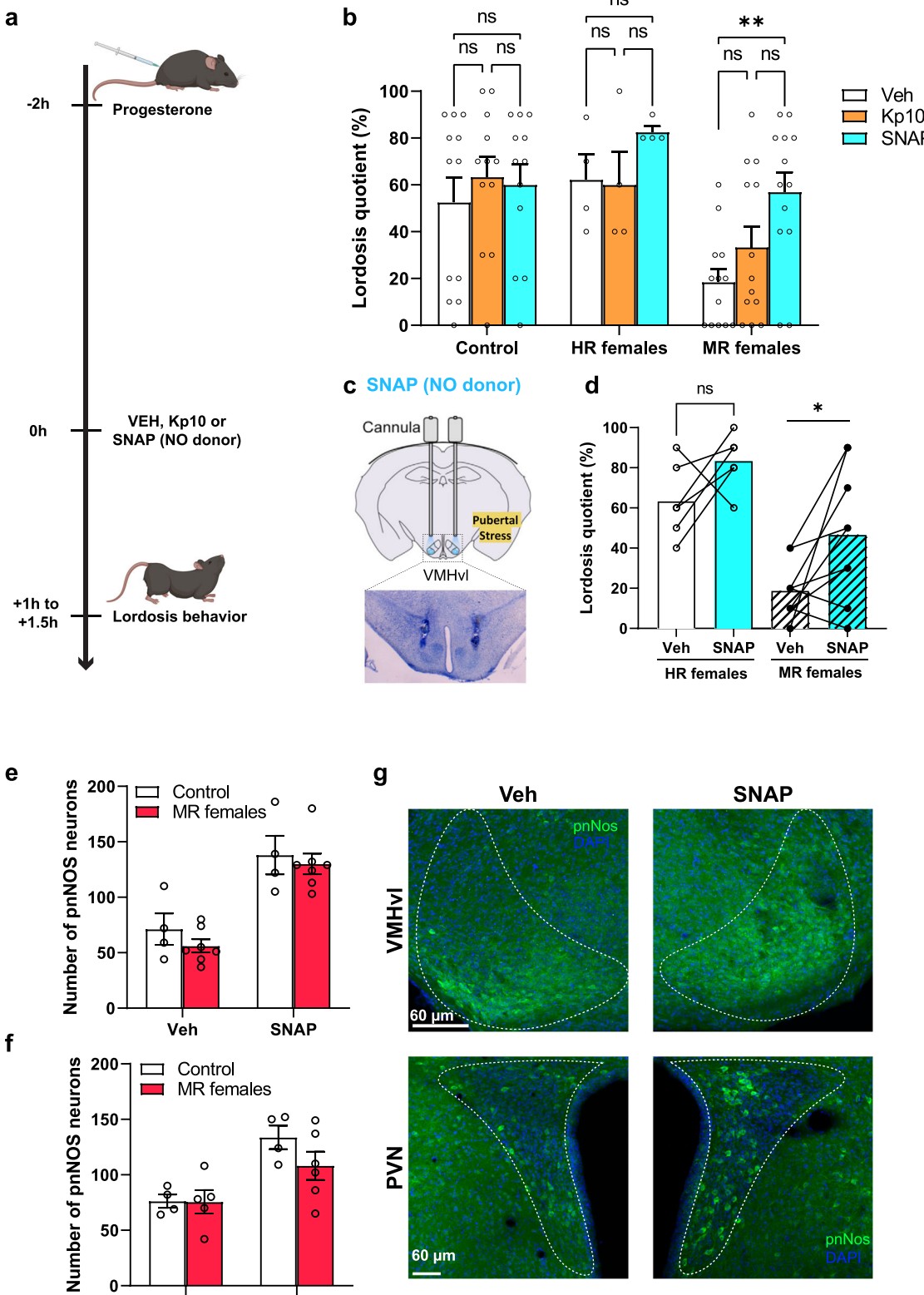

fluorescence responses were collected through the fluorescence minicube with an integrated photosensor on the RZ10x. The signals were demodulated at 1017.25 Hz. This process allowed dissociating the GCaMP6-dependent signals from the calcium-independent variations resulting from autofluorescence and motion artifacts. Data were recorded using Synapse Essentials software (Tucker Davis Technologies).

To measure the activity of the VMHvl nNOS neurons during lordosis behavior, females were connected to the patch cable and habituated to the recording setup. Five minutes later, the stimulus male was introduced, and lordosis behavior was scored for 10 min. Video scoring was performed offline to extract manually the time-stamps during which the male was introduced, and the female expressing lordosis or rejection behavior. All females were submitted

**Fig. 9 | Nitric oxide (NO) supplementation increased the number of phosphorylated NOS neurons and stimulated lordosis behavior in minimally receptive (MR) females. a** Timeline of vehicle, SNAP or kisspeptin (Kp10) Treatment. Figure created partially with BioRender.com. **b** Effect of subcutaneous treatment with kisspeptin or SNAP (NO donor) on lordosis behavior in control ($n = 12$), highly receptive (HR) females ($n = 4$) and MR females ($n = 13$). RM two-way ANOVA; Group effect: $F_{(2,26)} = 4.173$, $p = 0.027$; treatment effect: $F_{(1.747,45.42)} = 5.992$, $p = 0.007$; interaction between the two factors: $F_{(4,52)} = 2.163$, $p = 0.086$. ns: not significant, $**p = 0.003$; Bonferroni's multiple comparisons test. **c** Schematic representation of the infusion cannula into the VMHvl and the histological verification of the cannulas placement. Figure partially created with BioRender.com. **d** Effect of SNAP infusion into the VMHvl on lordosis behavior in HR females compared to MR females. RM two-way ANOVA; Group effect: $F_{(1,13)} = 24.14$, $p = 0.0003$; treatment effect: $F_{(1,13)} = 8.646$, $p = 0.0115$; group

× treatment effect: $F_{(1,13)} = 0.229$, $p = 0.64$. $*p < 0.0362$; Bonferroni's post hoc test. $n = 6$ for HR females, and $n = 9$ for MR females. **e** Number of pnNOS neurons in the VMHvl following vehicle or SNAP subcutaneous administration. Two-way ANOVA; group effect: $F_{(1,18)} = 1.107$, $p = 0.3067$; treatment effect: $F_{(1,18)} = 40.98$, $p < 0.0001$; interaction: $F_{(1,18)} = 0.1047$, $p = 0.75$. Control treated with vehicle ($n = 4$), control treated with SNAP ($n = 4$), MR females treated with vehicle ($n = 7$), MR females treated with SNAP ($n = 7$). **f** Number of pnNOS neurons in the PVN in vehicle or SNAP treated female mice. Two-way ANOVA; factor group: $F_{(1,15)} = 1.34$, $p = 0.26$; treatment: $F_{(1,15)} = 15.85$, $p = 0.0012$; group × treatment: $F_{(1,15)} = 1.214$, $p = 0.287$. $n = 4$ for the control treated with vehicle or SNAP, $n = 5$ for MR females treated with vehicle, $n = 6$ for MR group treated with SNAP. **g** Representative confocal images showing pnNOS expression in vehicle and SNAP treated females. All data are presented as mean ± SEM. Adjustments were made whenever multiple comparisons test is used. Source data are provided as a Source Data file.

to two lordosis pre-tests before undergoing the fiber photometry recording.

To record the activity of the VMHvl nNOS neurons in response to odors, mice were connected to the fiber photometry patch cable and introduced into the recording aquarium with clean sawdust bedding. To acclimate the subject to the recording environment, they were allowed to explore for 3 to 5 min before the photometry recording started. A cotton swab was dipped into an Eppendorf containing either male urine, estrous female urine, amyl acetate or water and presented to the subject. Odors were presented 3 to 4 times during each recording session, in a random way and spaced by a duration of 60 to 90 seconds. Video recordings were used to identify and extract timestamps of the cotton swab investigation. The urine used as stimulus was collected from a gonadally intact male or estrous female and stored in −20 °C until the day of use, while amyl acetate (Merck, 818700) was diluted to a concentration of 1:1000 (v/v)[46]. Data analyses were performed using a Python toolbox[55], while plotting was carried out with Prims 9 (GraphPad). Raw data was extracted and then the isosbestic channel was fitted to the GCaMP6s channel using a least square linear regression. Delta F/F was calculated by subtracting the fitted isosbestic channel from the GCaMP6s channel and then dividing by the fitted control channel. For peri-event time histograms (PETH), time zero was set to the start of the behavior or the event of interest, and baseline value of fluorescence was calculated during the five seconds time window prior to the onset of each event. The peak delta F/F was calculated within 10 seconds following the introduction of the male when tested for lordosis behavior or the presentation of the cotton swab in the odor test, while we used a 5-second window for lordosis and rejection behavior.

## Elevated plus maze

The evaluation of anxiety-like behavior was performed in the elevated plus maze apparatus, which consisted of two closed arms (38 cm long × 5 cm wide × 15 cm high) and two open arms (38 cm long × 5 cm wide × 2 cm high)[56,57]. The maze was raised from the floor at a height of 80 cm. Before each subject was introduced, the maze was cleaned with a 70% ethanol solution. The EPM test was performed under red light illumination on gonadally intact females at the stage of proestrus/estrus of the reproductive cycle. The ratio of open arm entries (open arms entries/ total entries) and the ratio of time spent in the open arm (open arms time/total time) were used as indices of anxiety. The results are expressed as a percentage (ratio × 100).

## Forced swim test

Depression-like behavior was evaluated on gonadally intact females. mice were allowed to swim in a 2-liter glass beaker filled with water up to 75% of its capacity. The water temperature was maintained at 25 °C. The behavior was videotaped for 6 min, with the first two minutes removed from the analysis because most mice are very active at the

beginning of the test and potential differences can be obscured during the first 2 min[56,58]. The behavioral analysis consisted of quantifying the duration of immobility. The subject was considered immobile when floating and making movements only required for keeping the head above water. At the end of the test, mice were dried before returning to their home cage.

## Lordosis test

All lordosis tests were conducted in a Plexiglas aquarium (32.5 cm long × 17.5 cm wide × 18.5 cm high) containing clean sawdust bedding. All females were primed with progesterone (500 µg) 2 to 3 h prior to each testing session. A sexually experienced male was placed in the aquarium and allowed to adapt for 10 min before introducing a hormonally primed stimulus female. Once the male started showing sexual behavior, the stimulus female was replaced by the experimental one and lordosis behavior was scored for 10 min or until the female had received 10 mounts, whatever came first. If the male did not mount the experimental female, she was transferred to a different aquarium with a different male. A lordosis quotient (LQ) was calculated by dividing the number of lordosis responses displayed by the female subjects by the number of mounts received (×100), whereas sexual rejection was calculated by dividing the number of rejection responses by the female subjects by the number of mounts received (×100).

## Sexual preference

Sexual preference was evaluated using the mate preference test in a box (60 cm long × 30 cm wide × 30 cm high) divided into three compartments interconnected by small doors. The lateral compartments contained a small section for the stimulus animals (10 cm wide × 20 cm long). One day before the behavioral experiment, mice were habituated to the apparatus by placing them in the middle compartment with free access to the other compartments for 10 min. No stimulus animals were used during the habituation session. At the day of testing, a gonadal intact male and an estrous female were placed in the small sections of the lateral compartments with their bedding to boost the olfactory stimuli. The experimental subject was placed in the central compartment and videotaped for 10 min. The time spent in each compartment was manually scored, and the preference score was calculated using the following formula: time spent in the male compartment−time spent in the female compartment/total time spent in both compartments. A positive score implicated a preference directed towards the male, while a negative value indicated a preference directed to the female. All experimental females were primed with progesterone 2 to 3 h as described above before testing for sexual preference.

## Preparation of male soiled bedding for odor stimulation

Gonadally intact males ($n = 15$) were housed individually in clean cages containing fresh sawdust. Soiled bedding was collected 72 h later and directly used as olfactory stimulus for experimental females.

## Bedding Exposure

To induce Fos expression in brain areas implicated in sexual behavior, females were exposed to either clean or male soiled bedding collected as indicated above. Forty-eight hours before bedding exposure, females were placed on clean sawdust bedding and housed in a separate housing unit that did not include the males. During the day of bedding exposure, females received a progesterone injection (500 µg) to induce behavioral estrus. Two to 3 h later, females were transferred into a cage that contained freshly collected male-soiled or clean bedding. Then ninety minutes later, brains were collected, as detailed below.

## Tissue collection and histology

Animals were deeply anesthetized with Euthasol (400 mg/Kg) and transcardially perfused with saline followed by ice-cold 4% paraformaldehyde (PFA). After collecting the brains, they were kept overnight in 4% PFA for post-fixation, then cryoprotected in 30% sucrose for 24 h at 4 °C. The brains were cut in a Fisher Scientific NX70 cryostat in the coronal plane and collected in 4 series (40 µm thick), placed in antifreeze solution, and stored at −20 °C until being used for immunohistochemistry.

## Double label Kisspeptin/Fos immunohistochemistry

Immunolabeling for Fos was performed first then followed by kisspeptin. Briefly, sections were incubated successively in 0.1% $NaBH_4$ (15 min), 3% $H_2O_2$ (30 min), 10% normal goat serum (1 h) and then 48 h at 4 °C in a guinea pig polyclonal anti-c-Fos antibody (1:1000; Synaptic systems, 226005). Then they were incubated for 1 h in a goat anti-guinea pig biotinylated antibody (1:1000; Vector Labs, BA-7000) followed by ABC complex (1:800; Vector Labs, PK-6100) for 1 h. The peroxidase was visualized with 3,3′-diaminobenzidine to give a brown precipitate. Then, the residual peroxidase activity was blocked in $H_2O_2$ (3%), followed by a 1-h incubation in 5% normal goat serum. Sections were further processed for kisspeptin using a rabbit polyclonal anti-kisspeptin 10 antibody (1:10000; Alain Caraty, INRA, France) for 48 h at 4 °C. Brains sections were then incubated in a goat anti-rabbit biotinylated antibody (Jackson immuneResearch, 111-065-003) followed by the ABC complex for 1 h, and the peroxidase was visualized by a modified DAB to give a blue precipitate (Vector Labs, SK-4700). Sections were mounted, cleared with xylene and cover slipped with Eukitt mounting medium.

## Double-label fluorescence immunohistochemistry

Double labeling for nNOS/Fos, or nNOS/GFP(GCaMP) was performed using standard procedures. Primary antibodies were used at the following concentrations: polyclonal rabbit anti-nNOS (1:500; Thermo-Fisher, 61-700), polyclonal guinea pig anti-c-Fos (1:500; Synaptic Systems, 226005) and FluoTag-X2 anti-GFP labeled with Atto488 (1:500; NanoTag Biotechnologies GmBH, N0304-At488). Brain sections were washed with PBS, incubated in 0.1% triton X-100, 5% normal goat serum in PBS for 1 h at room temperature, followed by 48 h incubation at 4 °C with a mixture of primary antibodies in the blocking solution. Following washes with PBS, brain sections were incubated with the secondary antibodies at room temperature for 1h30. Alexa Fluor-546 goat anti-rabbit (A-11010) and Alexa Fluor-488 goat anti-guinea pig (A-11073) were all purchased from ThermoFisher Scientific and used at 1:500 dilution in PBS. Sections were mounted on the same day and cover slipped with Aqua-poly-mount (Polysciences Inc., 494333).

## Immunofluorescence with antibodies from the same host species

To perform immunofluorescence with two primary antibodies raised in the same host species, we used a slightly different protocol compared to the one described above. Briefly, coronal sections were blocked in PBS with 5% normal goat serum and 0.3% triton X-100 for 1 h at room temperature before incubation with rabbit anti-nNOS (1:500) in the same blocking solution for 48 h at 4 °C. sections were washed in PBS and incubated in the secondary antibody Alexa Fluor 488 goat anti-rabbit (1:500) for 2 h at room temperature in 5% normal goat serum. After extensive washing, sections were incubated for 1 h in 10% normal rabbit serum (011-000-120, Jackson ImmunoResearch) and 0.3% Triton X-100, followed by 1 h incubation in unconjugated FAB goat anti rabbit (30 ug/ml; 111-007-003, Jackson ImmunoResearch). These steps are done to saturate open binding sites in the first primary and first secondary antibody so that they cannot be captured by the next antibodies. Sections were thoroughly washed in PBS and blocked in 10 normal goat serum for 30 min, then incubated for 24 h at 4 °C in one of the following primary antibodies; rabbit anti-GR (1:200; 24050-1-AP, Proteintech), rabbit anti-estradiol receptor alpha (Erα) (1:2000; 06-935, Millipore), rabbit anti-progesterone receptor (PR) (1:100; A0098, Dako). After washes, slices were incubated for 2 h in the secondary antibody Alexa 594 goat anti-rabbit (1:500; 111-585-003, Jackson ImmunoResearch). Sections were then incubated for 10 min in DAPI then mounted and cover slipped the same day using Aqua polymount.

## Immunofluorescence for pnNOS

To measure the phosphorylation of nNOS neurons, female mice were subcutaneously injected with either vehicle or a mixture of SNAP/Bay-41-22711 as indicated above and were perfused 15 min later. Brain sections were washed in PBS and incubated in 5% normal goat serum diluted in PBS containing 0.3% of triton X-100 for 1 h, followed by overnight incubation with a rabbit anti-phospho-nNOS (Ser1417) antibody (1:500; PA1032, ThermoFisher) at 4 °C. Sections were then incubated with an Alexa Fluor 488 goat anti-rabbit at room temperature for 2 h. Slices were mounted the same day and cover slipped with aqua-poly mount (Polysciences Inc., 494333).

## Imaging analysis and quantification

Images of visible staining were taken using an Olympus slide scanner (SlidViewer-VS200), while sections with fluorescence staining were imaged with the Zeiss LSM980 confocal microscope. Quantification of kisspeptin/Fos co-expression was done manually. Both hemispheres of 3 sections per animal were counted along the RP3V. Quantifications of nNOS/GFP, nNOS/Fos, nNOS/PR, nNOS/ER, nNOS/GR or nNOS were also done manually on both hemispheres in one representative section per subject between bregma −1.58 and −1.70 mm for the VMHvl, and between bregma −0.82 and −0.94 for the PVN, according to the mouse brain atlas[59].

Numbers of Fos immunoreactive neurons were counted in several brain nuclei known to be involved in olfactory processing and sexual responses[19]. Quantification was performed bilaterally on one representative section using the particles counter plugin of image J (Ver. 1.53 NIH, USA). Images were converted to binary and manually thresholded to discriminate the labeling from the background. Then the number of Fos-expressing neurons was counted automatically with the particle analyzer command in the selected region of interest.

## Statistics

Statistical analysis was performed using GraphPad Prism V9 software. Data were assessed for normality with the Shapiro−Wilk test. Mann−Whitney $U$ test was used to compare two experimental groups in which unpaired data were not normally distributed. For unpaired samples that were normally distributed, we used two-tailed Student's $t$ test. Paired data were analyzed with the Wilcoxon signed-rank test, paired $t$ test or repeated measure one-way ANOVA. Group analysis with two variables was carried out using two-way ANOVA. When appropriate, ANOVA was followed by Bonferroni or Dunn's multiple comparisons test. Statistical significance was set at $p < 0.05$. For statistical

analysis of data presented in Fig. 4 and onward, pubertally stressed females were divided into two subpopulations according to their sexual receptivity. Minimally receptive (MR) females are defined as subjects with an average lordosis quotient that is equal or below 30%, while highly receptive (HR) females are subjects with an average lordosis quotient above 30%.

## Reporting summary

Further information on research design is available in the Nature Portfolio Reporting Summary linked to this article.

## Data availability

The data generated or analyzed during this study are included in the published article and its supplementary information files and are available from the corresponding author upon request. Source data are provided with this paper.

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

## Acknowledgements

This work was supported by research grants from the Fonds National de la Recherche Scientifique (FNRS; (J.0008.21) to J.B. and (1.B.035.21F) to Y.B.) and the University of Liège (FRS-S-SS-20/27) to J.B. Y.B. is a FNRS Chargé de Recherches and J.B. is a FNRS research director. We thank Mike Baum and Jacques Balthazart for their comments on an earlier version of this manuscript.

## Author contributions

Y.B. and J.B. designed the study. Y.B. collected the data, undertook the statistical analyses, and wrote the first draft of the manuscript. Both authors reviewed and edited the manuscript.

## Competing interests

The authors declare no competing interests.
