## [Peer Review File · Nature Communications]

Stress during pubertal development affects female sociosexual behavior in miceREVIEWER COMMENTS

Reviewer #1 (Remarks to the Author):

This is a very nice study investigating the mechanism underlying pubertal stress-induced reduction in sexual behavior in mice. The authors found that pubertal stress permanently disrupted sexual performance without affecting sexual preference. Pubertal stress also reduced expression and activation of neuronal nitric oxide synthase (nNOS) in the ventrolateral part of the ventromedial hypothalamus (VMHvl). Fiber photometry revealed that VMHvl nNOS neurons are strongly responsive to male olfactory cues with this activation being substantially reduced in pubertally stressed females. Finally, treatment with a NO donor partially restored sexual performance in pubertally stressed females. This study is well designed and well written. It provides novel insights into the involvement of VMHvl nNOS in the processing of olfactory cues important for the expression of female sexual behavior. The authors also found that exposure to stress during puberty disrupts the integration of male olfactory cues leading to reduced sexual behavior. I really liked this work and I just have minor comments.

1. The authors state that pubertal stress did not affect sexual preference, but no sexual preference test was actually carried out in this study. I would recommend describing these findings differently without referring to sexual preference.
2. Finally, there are several typos in the text and I also noted one in Figure 2. I would commend that the authors go through the manuscript to correct the typos.

Reviewer #2 (Remarks to the Author):

In this study the authors evaluate the role of the neuronal nitric oxide synthase-expressing neurons located in the ventromedial hypothalamus in the control of female sexual behavior in conditions of chronic stress during the pre- and peri-pubertal period.

The authors initially assess how chronic stress, initiating prior to puberty, can affect female estrous cyclicity and sexual behavior. They show that following their stress protocol, females demonstrate impaired estrous cyclicity and decreased lordosis quotient comparing to the non-stressed control group. Interestingly, authors further set a threshold in the average lordosis quotient that allowed them to reveal the existence of two subpopulations of i) minimally receptive and ii) highly receptive females. Eventually, the authors checked that the differences in the lordosis quotient do not reflect olfactory-discrimination deficits (measuring the male preference score) or an increased anxiety/ depression-like behavior.

The authors then provide new insights into the mechanistic details underlying the impaired sexual behavior of the stressed females. For that, authors exposed stressed females to male bedding -a protocol known to activate specific subsets of neurons, like the POA kisspeptin neurons, involved in sexual behavior- revealing that stress leads to a decrease in the activation of the VMH nNOS neurons in response to the introduction of the male olfactory cue. Authors performed in vivo calcium recordings of the VMH nNOS neurons, allowing them to record the activity of the neuronal population in question in mice connected to the fiber photometry setup, in response to odors, and during the lordosis or rejection behaviors, and also assessed the implication of sex steroids (progesterone and estrogen) by distinguishing three groups: i) OVX, ii) OVX plus estradiol and iii) OVX plus estradiol and progesterone. Finally, authors revealed a partial rescue of the lordosis behavior in the stressed females upon administration of a NO donor into the VMH. It is quite interesting that the authors revealed that nNOS neurons are not directly involved in the expression of lordosis or rejection behavior, in the absence of an olfactory cue. Eventually authors present hormonal data showing a reduced corticosterone/DHEA ratio in the

minimally receptive group of the stressed females in response to exposure to a stress stimulus (elevated plus maze).

Altogether, the study contains novel information with an interesting experimental design, including the highly informative *in vivo* calcium recordings, and a solid assessment of lordosis, rejection and olfactory discrimination, techniques that the authors seem to master. To this reviewer's knowledge, even though literature has already highlighted the importance of VMH nNOS neurons in the control of female sexual behavior (10.1016/j.neuropharm.2021.108762, 10.1073/pnas.2203503119) there was no prior study demonstrating a direct involvement of VMH nNOS activity. More importantly, even though studies have suggested that VMH activity can impact on the regulation of emotional state (10.1016/j.celrep.2017.11.089, 10.1016/j.neuron.2014.12.025), no prior studies have provided evidence of a stress-induced modulation of VMH neuronal activity (especially during postnatal maturation), impacting eventually on sexual behavior. Overall, the work makes novel suggestions that could potentially be of a high importance for the evolvement of our knowledge on the effect of stress during young age on the future sexual behavior, hence opening up to new potential therapeutic interventions in populations facing high stress during their youth (e.g. aggression, sexual abuse, physical abuse etc.). However, the authors sometimes do not investigate in depth their findings, making assumptions, that might be indeed logical, yet result to the manuscript giving unrightfully the impression that there are many data acquired but not presented. Answering questions naturally raised while reading the manuscript (some of them the authors ask themselves in their discussion) would strengthen the work of the authors and solidify their novel findings, while it would definitely add up to the novelty of their work.

My specific comments are therefore:

1. Already during the introduction authors should become a bit more specific concerning what they consider as the period of "brain development" (line 34) and/or the key period for stressed-induced adaptations in the sexual behaviour. The ref. 5-8 refer to the importance of pubertal period (6 weeks of postnatal age), i.e., when females have already started demonstrating the first ovulation (6 weeks actually corresponds more to the very end of the peripubertal period, beginning of adulthood). Yet, lines 40-46 refer to the importance of "pre-pubertal" stage (prior to P25), i.e., when females are not yet sexually mature (P15-P25 would be later infantile, early juvenile stage). In addition, the authors refer to both pre- and post- pubertal periods being impacted by the chronic social isolation (line 46), and few lines later (in line 55) the authors refer to the time "during puberty"- yet, both Ruscio et al., and Kercmar et al. (studies that the authors cite in that statement), induced chronic stress in pre-pubertal animals till early adulthood. We would expect that stress pre- or at/post-puberty wouldn't affect the same pathways e.g., ovarian hormone patterns, or VNO development (which could itself impact on sexual and aggressive mouse behaviour...). Intermingling different ages creates confusion and naturally raises the questions of whether it is really the stress impacting on puberty (and subsequently all related developmental processes) or if it's rather the sexual (in)experience impacting on the "wiring" of neuronal networks and hence affecting sexual behaviour. This for example, is something that the authors very interestingly discuss in lines 376-385. I would hence suggest authors to be clearer about what they want to suggest as a crucial period, allowing us to better understand why they chose this experimental protocol of chronic stress (that doesn't only affect pubertal period but starts from late infantile) and the possible consequences of this protocol.

2. In line with my comments on the introduction: The authors induced chronic stress in the female mice by single-caging them from the time of weaning (P21) till P50. Considering female mice demonstrate vaginal opening (an external sign of sexual maturation) around P30 and they have their puberty about 10days later did the authors check whether these phenomena were disrupted in the stressed animals? If the authors would like to suggest puberty is an important time for future sexual

behaviour it would be important for us to know whether puberty itself is impacted (especially since impairments in the timing of puberty have been extensively associated with impaired sexual function).

Also, estrous cyclicity is ofc an important readout of female reproduction, yet more information would be needed on the impact of early-chronic stress on the HPG axis. Did the authors check whether proestrus and hence ovulation (that is seen to occur sporadically), corresponded to a GnRH/LH surge? For example, nNOS deficient animals, even when in proestrus they fail to demonstrate an LH surge. Did the authors assess LH pulsatility in the stressed females? Are the differences in the estrous cycle stemming from an alteration in the sex hormone response? This latter point is also quite important considering that, as the authors rightfully mention, prepubertal estradiol is needed for the development of female sexual behaviour did the authors check the estradiol and progesterone levels in the prepubertal stressed animals? Considering that both estradiol and progesterone are increased in the end of minipuberty and further increased at puberty (P40) it would be interesting to assess their levels in their experimental protocol. Indeed, this is something the authors point out in line 322. I feel this should be reported in their work.

3. The authors observe a striking decrease in the number of Fos positive neurons in the VMH, corresponding to a decrease in the nNOS neurons expressing Fos, stemming from an overall decrease in the number of nNOS neurons in the stressed animals. These results are extremely interesting. Considering that 1) stress can impact responses to sex hormones and 2) previous studies have identified the expression of ER α and Pr by nNOS neurons of the VMHv1 (Chachlaki et al., 2017 and Silva et al., 2022), whose activity depends on sex steroids, did the authors check whether pubertal stress alters the ER α and PR expression profile in nNOS VMH neurons or whether it modifies the phosphorylation of nNOS enzyme? It would significantly improve the strength of the manuscript if authors were presenting data on a possible deregulation of those expression patterns in the MR females.

This is something the authors themselves point out in their discussion (lines 353-358).

The possible modulation of nNOS by sex steroids is discussed by the authors in several occasions, yet conclusions seem a bit rushed. For example:

Line 314: Could authors explain this statement? If they refer to the study from Kercmar et al., in that study ovx females were supplemented only with estradiol (not P) in late adulthood. Hence, this doesn't exclude that defect arises from abnormal pubertal sex hormone levels subsequently disrupting neuronal pathways (e.g. activation of nNOS neurons). Also, the authors themselves demonstrated that supplementation with E2 coupled with P was able to rescue nNOS activity in response to male odor...

Line 322-26: The authors use the example of kisspeptin neurons however they have focused their hypothesis on nNOS neurons. In contrast to what may happen with RP3V kisspeptin neurons, nNOS activity has been shown to be highly sensible to pre-pubertal and post-prepubertal estrogen levels as well as ER α activity in both sexes (Chachlaki et al., 2022, Delli et al., 2023) and the authors themselves showed a reduction in nNOS neuron number as well as Fos expression in MR females. Maybe authors care to re-phrase or comment.

4. The authors conclude that Progesterone is modulating the activation of nNOS neurons in response to olfactory cues. This is also stated in the discussion, line 361. Yet, remains an assumption based on their results upon OVX vs OVX+E2 vs OVX+E2+P. It would be important (and quite easy) for the authors to assess activity of nNOS neurons upon P administration, as well as reveal some leads on how this circuit could be impaired in the MR group (if this is the case). Especially because P and E2 have been proposed to work in synergy to facilitate sexual receptivity in ovariectomized mice and have been seen to synergistically alter nNOS-ir in other hypothalamic nuclei (<https://doi.org/10.1016/j.brainres.2011.06.017>, <https://doi.org/10.1016/j.cell.2019.10.025>).

5. The authors demonstrate that microinfusion of a SNAP, a NO donor into the VMHvl rescues sexual behavior in females. This is not the first time that NO supplementation has been shown to normalise lordosis behavior. Would be interesting, and significantly add on the novelty of the paper, if the authors explored how this is managed. For example, NO release is known to modulate NO production (and nNOS activity) in an auto-regulation loop: could SNAP act by re-establishing normal nNOS activity pattern that the authors identify as impaired?

6. I would suggest the authors to present the data suggesting an alteration of the HPA axis in the beginning of the manuscript, rather than in the end. Seems the first question you ask as a reader considering that in the introduction the authors already set the ground on how stress can impact sexual function and sexual behaviour: i.e., is HPA axis impacted by their protocol and is this leading to an impaired HPG axis function/development?

Besides, the authors demonstrate the existence of an impairment specifically in the MR group, yet they chose to present data on the ratio, rather than the differences in the concentration of DHEA and corticosterone in MR vs HR vs control females. Since the authors very early identified this dichotomy in the response of the pubertally stressed animals it would be important to demonstrate clearly those group differences. If DHEA and corticosterone levels are not different in the MR vs HR vs control females what does a difference in the ratio signifies (physiologically)? Authors are invited to comment on this in their discussion.

7. Effect of progesterone and corticosteroids on NO producing cells has been suggested to differ according to the neural location studied. For example, studies in the ewe have reported that estradiol results in a decrease of GR expression in the nNOS neurons of the VMHvl (<https://doi.org/10.1095/biolreprod.102.004648>). In parallel, chronic mild stress and CORT increase have been associated with increased nNOS in the hippocampus, hence pathogenic amounts of NO, leading to NO excitotoxicity (nitrosative stress).

It would be relevant for authors to assess GR expression in the VMH/ PVN and its colocalization with nNOS... Would SNAP/ L-NAME alter GR expression? Indeed, lack of information on glucocorticoid receptor expression by nNOS neurons is something the authors discuss (lines 347-348). If the authors aim to create a link between stress and sexual behaviour with the identified role of VMH nNOS neurons, and given the state-of-the-art techniques the authors are presenting here, the co-expression of glucocorticoid receptors is sth that could be easily addressed in this manuscript, rather than remaining an open question.

Minor comments:

Line 61: I would invite the authors to refer also to the latest study from Silva et al., 2022 demonstrating the functional relevance of VMH nNOS neurons in lordosis behaviour

Line 73: I would invite authors to refer to relevant rodent studies suggesting the importance of puberty and sex steroids on HPA axis activity and glucocorticoid hormones, for example the early studies from the Lightman group

Line 159: I would invite authors to also ref to figure 2e that also demonstrates this decrease in the fos signal

Line 162: Indeed, nNOS neurons of the VMH are highlighted as key for lordosis behaviour by several studies, including a recent study applying a chemogenic approach to specific target this population and demonstrate that in the absence of nNOS activity in the VMH lordosis quotient is decreased and kisspeptin is unable to rescue it (Silva et al., 2022).

Line 334: This phenotype has been also shown to occur when there is deficient nNOS signaling, (Chachlaki et al., 2022), maybe more relevant in view of what the authors propose.

Lines 439-442: I would invite the authors to state the age in which estrous cyclicity was assessed. Early or late adulthood?

Reviewer #3 (Remarks to the Author):

This interesting study addresses the long-term impact of peripubertal stress (induced by social isolation) on female sexual behaviour in adulthood, as assessed by lordosis in response to male stimulus. By a combination of expression and functional studies, including fibre photometry, the authors propose a novel pathway, involving nNOS neurones in the VMN as key component for a lordosis behaviour, but not sex preference, which is sensitive to pubertal stress. The concept that pubertal stress can persistently affect behavioural traits in adulthood had been established before, but the present study nicely dissects out the pathway whereby this can impair sexual behaviours, with a primary impact distal to RP3V Kisspeptin neurones, on VMN nNOS neurones. While the observations are of interest, there are some issues that would benefit from further elaboration by the authors

Major Comments

1. The issue of the uneven manifestation of minimal vs. high receptivity among the animals of the two groups (control and stressed) is very interesting but actually not addressed in the current study. The authors refer to cellular heterogeneity, but it is not clear to this referee if such heterogeneity lies on nNOS neurones, or upstream or down-stream elements of the proposed pathway. The fact that 40% of stressed animals remained highly receptive is interpreted as manifestation of a majority of nNOS neurones being refractory to the stress manipulation in this 40% but not in the remaining subgroup? Were other forms of stressors tested, even preliminarily, to ascertain whether stronger or weaker stress stimuli may bring different results?
2. Previous data have documented that kisspeptin output from RP3V neurones has a major role in lordosis behaviour. While according to c-fos data, these neurones are not apparently affected by the stressor, would it be possible that other parameters, such as Kiss1 mRNA expression, or more interestingly, activity patterns (as measured by fibre photometry) might have been affected, leading to impairment of activation of nNOS neurones in the VMN? This referee understands conducting fibre photometry in another set of neurones, as RP3V Kisspeptin neurones, is not trivial, but might help to clarify the pathway. Did the authors consider this possibility?
3. In the same line, the interplay with male odours is very interesting, but the proposed pathway is unclear. Comparison between nNOS vs. Kisspeptin neuronal activity in response to odours might help to delineate which neurones are primary responsive to these stimuli and altered in response to stress.
4. Were the effects of peripubertal vs. adult stress compared regarding the proposed pathway? In order to define differences in the plasticity of the circuits, it might be important to check whether similar responses are obtained or not after similar stress protocols applied in adulthood.
5. While the paper is focused on behaviours, did the authors consider the possibility to assess the impact of the stress protocol on reproductive hormonal profiles, such as endogenous progesterone and LH levels? This might be relevant from a mechanistic perspective, since an impairment of ovulatory function may lead to defective progesterone secretion (e.g., no or lower number of corpora lutea in stressed animals). In other words, while replacement experiments suggest that a central component is in place, the apparent progesterone dependence suggests that perturbations of the neuroendocrine profiles caused by pubertal stress might contribute to the observed phenomena.
6. Pharmacological experiments with the NO donor are interesting. Did the authors consider the possibility to compare SNAP results with those of kisspeptin administration in control vs. stressed

animals? No matter what the results are (if SNAP and kisspeptin do the same or not), this might be informative, especially considering that nNOS VMN neurones are responsive to kisspeptin.

7. In the discussion of the data, the authors tend to compare current results on impaired lordosis with low sexual desire in women. I would suggest this connection is further supported by previous literature, as low sexual desire in women may result from multiple components. Interestingly, recent studies suggest a connection between kisspeptin and sexual desire in men, so that kisspeptin treatment might enhance sexual desire in patients suffering hyposexual desire. Do the authors consider there might be some connection with present findings?

Please find below our corresponding replies to the reviewers. Most changes have been included in the revised manuscript, and they are highlighted in green.

REVIEWER COMMENTS

Replies to Reviewer #1:

Reviewer # 1 has made 2 comments that have been addressed.

This is a very nice study investigating the mechanism underlying pubertal stress-induced reduction in sexual behavior in mice. The authors found that pubertal stress permanently disrupted sexual performance without affecting sexual preference. Pubertal stress also reduced expression and activation of neuronal nitric oxide synthase (nNOS) in the ventrolateral part of the ventromedial hypothalamus (VMHvl). Fiber photometry revealed that VMHvl nNOS neurons are strongly responsive to male olfactory cues with this activation being substantially reduced in pubertally stressed females. Finally, treatment with a NO donor partially restored sexual performance in pubertally stressed females. This study is well designed and well written. It provides novel insights into the involvement of VMHvl nNOS in the processing of olfactory cues important for the expression of female sexual behavior. The authors also found that exposure to stress during puberty disrupts the integration of male olfactory cues leading to reduced sexual behavior. I really liked this work and I just have minor comments.

POINT 1. The authors state that pubertal stress did not affect sexual preference, but no sexual preference test was actually carried out in this study. I would recommend describing these findings differently without referring to sexual preference.

REPLY TO POINT 1: We think that the reviewer has probably missed the sexual preference data in figure 1. They are included in Figure 1k and 1l.

POINT 2. Finally, there are several typos in the text, and I also noted one in Figure 2. I would commend that the authors go through the manuscript to correct the typos.

REPLY TO POINT 2. We thank the reviewer for pointing out the typos. We did our best to correct them.

Replies to Reviewer #2:

Reviewer #2 has raised multiple points that have been considered.

In this study the authors evaluate the role of the neuronal nitric oxide synthase-expressing neurons located in the ventromedial hypothalamus in the control of female sexual behavior in conditions of chronic stress during the pre- and peri-pubertal period.

The authors initially assess how chronic stress, initiating prior to puberty, can affect female estrous cyclicity and sexual behavior. They show that following their stress protocol, females demonstrate impaired estrous cyclicity and decreased lordosis quotient comparing to the non-stressed control group. Interestingly, authors further set a threshold in the average lordosis quotient that allowed them to reveal the existence of two subpopulations of i) minimally

receptive and ii) highly receptive females. Eventually, the authors checked that the differences in the lordosis quotient do not reflect olfactory-discrimination deficits (measuring the male preference score) or an increased anxiety/ depression-like behavior.

The authors then provide new insights into the mechanistic details underlying the impaired sexual behavior of the stressed females. For that, authors exposed stressed females to male bedding -a protocol known to activate specific subsets of neurons, like the POA kisspeptin neurons, involved in sexual behavior- revealing that stress leads to a decrease in the activation of the VMH nNOS neurons in response to the introduction of the male olfactory cue. Authors performed in vivo calcium recordings of the VMH nNOS neurons, allowing them to record the activity of the neuronal population in question in mice connected to the fiber photometry setup, in response to odors, and during the lordosis or rejection behaviors, and also assessed the implication of sex steroids (progesterone and estrogen) by distinguishing three groups: i) OVX, ii) OVX plus estradiol and iii) OVX plus estradiol and progesterone. Finally, authors revealed a partial rescue of the lordosis behavior in the stressed females upon administration of a NO donor into the VMH. It is quite interesting that the authors revealed that nNOS neurons are not directly involved in the expression of lordosis or rejection behavior, in the absence of an olfactory cue.

Altogether, the study contains novel information with an interesting experimental design, including the highly informative in vivo calcium recordings, and a solid assessment of lordosis, rejection and olfactory discrimination, techniques that the authors seem to master. To this reviewer's knowledge, even though literature has already highlighted the importance of VMH nNOS neurons in the control of female sexual behavior ([10.1016/j.neuropharm.2021.108762](https://doi.org/10.1016/j.neuropharm.2021.108762), [10.1073/pnas.2203503119](https://doi.org/10.1073/pnas.2203503119)) there was no prior study demonstrating a direct involvement of VMH nNOS activity. More importantly, even though studies have suggested that VMH activity can impact on the regulation of emotional state ([10.1016/j.celrep.2017.11.089](https://doi.org/10.1016/j.celrep.2017.11.089), [10.1016/j.neuron.2014.12.025](https://doi.org/10.1016/j.neuron.2014.12.025)), no prior studies have provided evidence of a stress-induced modulation of VMH neuronal activity (especially during postnatal maturation), impacting eventually on sexual behavior. Overall, the work makes novel suggestions that could potentially be of a high importance for the evolvement of our knowledge on the effect of stress during young age on the future sexual behavior, hence opening up to new potential therapeutic interventions in populations facing high stress during their youth (e.g. aggression, sexual abuse, physical abuse etc.). However, the authors sometimes do not investigate in depth their findings, making assumptions, that might be indeed logical, yet result to the manuscript giving unrightfully the impression that there are many data acquired but not presented. Answering questions naturally raised while reading the manuscript (some of them the authors ask themselves in their discussion) would strengthen the work of the authors and solidify their novel findings, while it would definitely add up to the novelty of their work.

My specific comments are therefore:

POINT 1. Already during the introduction authors should become a bit more specific concerning what they consider as the period of "brain development" (line 34) and/or the key period for stressed-induced adaptations in the sexual behaviour. The ref. 5-8 refer to the importance of pubertal period (6 weeks of postnatal age), i.e., when females have already started demonstrating the first ovulation (6 weeks actually corresponds more to the very end of the peripubertal period, beginning of adulthood). Yet, lines 40-46 refer to the importance of "pre-pubertal" stage (prior to P25), i.e., when females are not yet sexually mature (P15-P25 would be later infantile, early juvenile stage). In addition, the authors refer to both pre- and post- pubertal periods being impacted by the chronic social isolation (line 46), and few lines later (in line 55) the authors refer to the time "during puberty"- yet, both Ruscio et al., and

Kercmar et al. (studies that the authors cite in that statement), induced chronic stress in pre-pubertal animals till early adulthood. We would expect that stress pre- or at/post-puberty wouldn't affect the same pathways e.g., ovarian hormone patterns, or VNO development (which could itself impact on sexual and aggressive mouse behaviour...). Intermingling different ages creates confusion and naturally raises the questions of whether it is really the stress impacting on puberty (and subsequently all related developmental processes) or if it's rather the sexual (in)experience impacting on the "wiring" of neuronal networks and hence affecting sexual behaviour. This for example, is something that the authors very interestingly discuss in lines 376-385. I would hence suggest authors to be clearer about what they want to suggest as a crucial period, allowing us to better understand why they chose this experimental protocol of chronic stress (that doesn't only affect pubertal period but starts from late infantile) and the possible consequences of this protocol.

REPLY TO POINT 1: Puberty is defined as the transition to a mature reproductive state. It is a developmental process that starts with the first sign of ovarian "activation" and ends with the onset of the first estrous cycle (DOI: 10.1038/nn1326 ; 10.1530/eje.0.151u151). Therefore, puberty is an extended phase of development, and it is more accurate to use "pubertal development" to refer to it. Indeed, pubertal development in mice starts with the vaginal opening even though hormonal activation precedes it. In our study, we were interested in the effect of stress on all physiological changes that happen during the pubertal developmental period, rather than a specific age by itself. Besides, it is unlikely that the effect of social isolation starts already on the first day of isolation; it is a continuous and cumulative process. This is why we used a stress protocol that starts at weaning (P21) and ends 10 days before adulthood (P50).

Changes are made to the introduction (page 2) and to the title of the manuscript to reflect this.

POINT 2. In line with my comments on the introduction: The authors induced chronic stress in the female mice by single-caging them from the time of weaning (P21) till P50. Considering female mice demonstrate vaginal opening (an external sign of sexual maturation) around P30 and they have their puberty about 10days later did the authors check whether these phenomena were disrupted in the stressed animals? If the authors would like to suggest puberty is an important time for future sexual behaviour it would be important for us to know whether puberty itself is impacted (especially since impairments in the timing of puberty have been extensively associated with impaired sexual function). Also, estrous cyclicity is ofc an important readout of female reproduction, yet more information would be needed on the impact of early-chronic stress on the HPG axis. Did the authors check whether proestrus and hence ovulation (that is seen to occur sporadically), corresponded to a GnRH/LH surge? For example, nNOS deficient animals, even when in proestrus they fail to demonstrate an LH surge. Did the authors assess LH pulsatility in the stressed females? Are the differences in the estrous cycle stemming from an alteration in the sex hormone response? This latter point is also quite important considering that, as the authors rightfully mention, prepubertal estradiol is needed for the development of female sexual behaviour did the authors check the estradiol and progesterone levels in the prepubertal stressed animals? Considering that both estradiol and progesterone are increased in the end of minipuberty and further increased at puberty (P40) it would be interesting to assess their levels in their experimental protocol. Indeed, this is something the authors point out in line 322. I feel this should be reported in their work.

REPLY TO POINT 2. We did not check LH surge and pulsatility. However, new data about vaginal opening (Fig. 1a, Fig. 2b), levels of estradiol at P40 (Fig. 2a) and P60 (Fig. 2b) as well as levels of progesterone at P60 (Fig. 2c) are added to the manuscript. We found that stress has no apparent effect on the onset of puberty and the levels of ovarian hormones during puberty and in early adulthood.

POINT 3a. The authors observe a striking decrease in the number of Fos positive neurons in the VMH, corresponding to a decrease in the nNOS neurons expressing Fos, stemming from an overall decrease in the number of nNOS neurons in the stressed animals. These results are extremely interesting. Considering that 1) stress can impact responses to sex hormones and 2) previous studies have identified the expression of ER α and Pr by nNOS neurons of the VMHvl (Chachlaki et al., 2017 and Silva et al., 2022), whose activity depends on sex steroids, did the authors check whether pubertal stress alters the ER α and PR expression profile in nNOS VMH neurons or whether it modifies the phosphorylation of nNOS enzyme? It would significantly improve the strength of the manuscript if authors were presenting data on a possible deregulation of those expression patterns in the MR females. This is something the authors themselves point out in their discussion (lines 353-358).

The possible modulation of nNOS by sex steroids is discussed by the authors in several occasions, yet conclusions seem a bit rushed.

REPLY TO POINT 3a. We agree with the reviewer. New data about the effect of pubertal stress on other neuronal subpopulations in the VMHvl including ER α and PR expressing neurons are added (Fig. 8). Briefly, we found that pubertal stress specifically reduced the number of nNOS neurons without impacting PR, ER α or their coexpression with nNOS in the VMHvl indicating a specific reduction in the production of nNOS enzyme.

Additional results about the phosphorylation of nNOS neurons in the VMHvl and the PVN are included now as well (Fig. 9e-9g). Following SNAP administration, we revealed no difference in the number of phosphorylated nNOS (measured through pnNOS immunostaining) in the VMHvl and the PVN.

POINT 3b. For example: Line 314: Could authors explain this statement? If they refer to the study from Kercmar et al., in that study ovx females were supplemented only with estradiol (not P) in late adulthood. Hence, this doesn't exclude that defect arises from abnormal pubertal sex hormone levels subsequently disrupting neuronal pathways (e.g. activation of nNOS neurons). Also, the authors themselves demonstrated that supplementation with E2 coupled with P was able to rescue nNOS activity in response to male odor...

REPLY TO POINT 3b. Female mice in the study from Kercmar et al. 2014 (DOI: 10.3389/fnbeh.2014.00337) received an estradiol implant and were primed with progesterone 3 to 4 hours before the behavioral tests (please check page 2 in their methods). We believe that the reviewer might have missed this information in the work of Kercmar and colleagues.

In our study we found no effect of pubertal stress on sex hormones as well as on kisspeptin neurons which are known for being developed under the effect of pubertal estradiol (Clarkson et al . 2006). In addition, we found that supplementation with estradiol and progesterone in OVX females (OVX+E2+P4) was able to increase the activity of nNOS neurons in response to male odors only in the control group. Indeed, females that were exposed to pubertal stress, specifically MR females, expressed a significantly lower response to male odors when treated with E2+P4.

POINT 3c. Line 322-26: The authors use the example of kisspeptin neurons however they have focused their hypothesis on nNOS neurons. In contrast to what may happen with RP3V kisspeptin neurons, nNOS activity has been shown to be highly sensitive to pre-pubertal and post-prepubertal estrogen levels as well as ER α activity in both sexes (Chachlaki et al., 2022, Delli et al., 2023) and the authors themselves showed a reduction in nNOS neuron number as well as Fos expression in MR females. Maybe authors care to re-phrase or comment.

REPLY TO POINT 3c. We agree with the reviewer. According to the above-mentioned studies (Chachlaki et al. 2023; Delli et al. 2023), the activity of nNOS neurons has been shown to be sensitive to pubertal estrogen. However, this was reported only for the OVLTA nNOS population, which has been found to be mostly involved in modulating GnRH/LH secretion. To our knowledge, there is no study showing a direct involvement of estradiol in the development of the VMHvl nNOS neurons.

In our study, we have focused on the kisspeptin neurons in the RP3V and the nNOS neurons in the VMHvl because they have been shown to be directly involved in lordosis behavior (Hellier et al, 2018; Bentefour and Bakker, 2021). Since we observed no effect of pubertal stress on the levels of ovarian hormones and kisspeptin neurons, we hypothesize that the observed defect in the VMHvl nNOS neurons could be associated with serotonergic neurons, which are known to be highly sensitive to stress, and of which it has been reported recently to send projections towards the VMHvl (Ye et al, 2022; Sci. Adv.).

POINT 4. The authors conclude that Progesterone is modulating the activation of nNOS neurons in response to olfactory cues. This is also stated in the discussion, line 361. Yet, remains an assumption based on their results upon OVX vs OVX+E2 vs OVX+E2+P. It would be important (and quite easy) for the authors to assess activity of nNOS neurons upon P administration, as well as reveal some leads on how this circuit could be impaired in the MR group (if this is the case). Especially because P and E2 have been proposed to work in synergy to facilitate sexual receptivity in ovariectomized mice and have been seen to synergistically alter nNOS-ir in other hypothalamic nuclei (<https://doi.org/10.1016/j.brainres.2011.06.017>, <https://doi.org/10.1016/j.cell.2019.10.025>).

REPLY TO POINT 4. New data is now added about the activity of the VMHvl nNOS neurons in female mice treated with progesterone only (Supplementary Fig. s6). We found that progesterone alone is not enough to increase the activity of nNOS neurons in response to male olfactory cues, confirming that estradiol and progesterone act in synergy to modulate the neuronal activity and facilitate sexual receptivity.

POINT 5. The authors demonstrate that microinfusion of a SNAP, a NO donor into the VMHvl rescues sexual behavior in females. This is not the first time that NO supplementation has been shown to normalise lordosis behavior. Would be interesting, and significantly add on the novelty of the paper, if the authors explored how this is managed. For example, NO release is known to modulate NO production (and nNOS activity) in an auto-regulation loop: could SNAP act by re-establishing normal nNOS activity pattern that the authors identify as impaired?

REPLY TO POINT 5. New data about the effect of SNAP on the activity of the nNOS neurons is added (Fig. 9e, 9f, 9g). We analyzed the effect of NO supplementation on the

phosphorylation of nNOS neurons in both control and MR females. Our data show that, compared to the controls, there is no difference in the number of pnNOS neurons in MR females following SNAP treatment, suggesting that NO supplementation reestablished nNOS activity.

POINT 6. I would suggest the authors to present the data suggesting an alteration of the HPA axis in the beginning of the manuscript, rather than in the end. Seems the first question you ask as a reader considering that in the introduction the authors already set the ground on how stress can impact sexual function and sexual behaviour: i.e., is HPA axis impacted by their protocol and is this leading to an impaired HPG axis function/development? Besides, the authors demonstrate the existence of an impairment specifically in the MR group, yet they chose to present data on the ratio, rather than the differences in the concentration of DHEA and corticosterone in MR vs HR vs control females. Since the authors very early identified this dichotomy in the response of the pubertally stressed animals it would be important to demonstrate clearly those group differences. If DHEA and corticosterone levels are not different in the MR vs HR vs control females what does a difference in the ratio signifies (physiologically)? Authors are invited to comment on this in their discussion.

REPLY TO POINT 6: DHEA and CORT are known to have opposite actions, indicating that it could be very informative to calculate the CORT/DHEA ratio. For example, a slight increase in the level of CORT that is associated with a small decrease in the DHEA that are statistically not significant can be more apparent when calculated as a CORT/DHEA ratio. This has already been demonstrated in clinical studies investigating depression and anxiety.

Data about the HPA axis is now presented in the second figure. Unfortunately, we do not have enough subjects from the group of HR females to perform a proper statistical analysis (see the graph below). These data were actually collected before we noticed the presence of MR and HR females following exposure to pubertal stress. As an alternative, we performed correlations between the concentrations of the HPA axis hormones and sexual receptivity (supplementary fig s3). We found a significant correlation between baseline CORT/DHEA ratio and female sexual receptivity.

POINT 7. Effect of progesterone and corticosteroids on NO producing cells has been suggested to differ according to the neural location studied. For example, studies in the ewe have reported that estradiol results in a decrease of GR expression in the nNOS neurons of the VMHvl (<https://doi.org/10.1095/biolreprod.102.004648>). In parallel, chronic mild stress and CORT

increase have been associated with increased nNOS in the hippocampus, hence pathogenic amounts of NO, leading to NO excitotoxicity (nitrosative stress).

It would be relevant for authors to assess GR expression in the VMH/ PVN and its colocalization with nNOS... Would SNAP/ L-NAME alter GR expression? Indeed, lack of information on glucocorticoid receptor expression by nNOS neurons is something the authors discuss (lines 347-348). If the authors aim to create a link between stress and sexual behaviour with the identified role of VMH nNOS neurons, and given the state-of-the-art techniques the authors are presenting here, the co-expression of glucocorticoid receptors is sth that could be easily addressed in this manuscript, rather than remaining an open question.

REPLY TO POINT 7: Additional data about nNOS coexpression with the glucocorticoid receptor (GR) in the VMHvl (Fig. 4b, 4c, 4d) and the PVN (Fig 4f, 4g, 4h) are now added to the manuscript. We found no effect of pubertal stress on the subpopulation of neurons coexpressing nNOS and GR. Therefore, we believed that it was not necessary, for the current publication, to check whether SNAP/ L-NAME would alter GR expression.

Minor comments:

Line 61: I would invite the authors to refer also to the latest study from Silva et al., 2022 demonstrating the functional relevance of VMH nNOS neurons in lordosis behaviour.

⇒ **Reference added. Highlighted in page 3.**

Line 73: I would invite authors to refer to relevant rodent studies suggesting the importance of puberty and sex steroids on HPA axis activity and glucocorticoid hormones, for example the early studies from the Lightman group.

⇒ **References are now included. Highlighted in the first line of page 4.**

Line 159: I would invite authors to also ref to figure 2e that also demonstrates this decrease in the fos signal.

⇒ **Reference to the figure added. Highlighted in page 8.**

Line 162: Indeed, nNOS neurons of the VMH are highlighted as key for lordosis behaviour by several studies, including a recent study applying a chemogenic approach to specific target this population and demonstrate that in the absence of nNOS activity in the VMH lordosis quotient is decreased and kisspeptin is unable to rescue it (Silva et al., 2022).

⇒ **Reference added. Highlighted in page 8 as well.**

Line 334: This phenotype has been also shown to occur when there is deficient nNOS signaling, (Chachlaki et al., 2022), maybe more relevant in view of what the authors propose.

⇒ **Reference added. Highlighted in page 16.**

Lines 439-442: I would invite the authors to state the age in which estrous cyclicity was assessed. Early or late adulthood ?

⇒ **Estrous cyclicity is assessed in early adulthood, starting from P60. This information is now included in the methods (page 20).**

Replies to Reviewer #3:

Reviewer #3 has raised 7 points that have been fully addressed.

This interesting study addresses the long-term impact of peripubertal stress (induced by social isolation) on female sexual behaviour in adulthood, as assessed by lordosis in response to male stimulus. By a combination of expression and functional studies, including fibre photometry, the authors propose a novel pathway, involving nNOS neurones in the VMN as key component for a lordosis behaviour, but not sex preference, which is sensitive to pubertal stress. The concept that pubertal stress can persistently affect behavioural traits in adulthood had been established before, but the present study nicely dissects out the pathway whereby this can impair sexual behaviours, with a primary impact distal to RP3V Kisspeptin neurones, on VMN nNOS neurones. While the observations are of interest, there are some issues that would benefit from further elaboration by the authors.

Major Comments

POINT 1. The issue of the uneven manifestation of minimal vs. high receptivity among the animals of the two groups (control and stressed) is very interesting but actually not addressed in the current study. The authors refer to cellular heterogeneity, but it is not clear to this referee if such heterogeneity lies on nNOS neurones, or upstream or down-stream elements of the proposed pathway. The fact that 40% of stressed animals remained highly receptive is interpreted as manifestation of a majority of nNOS neurones being refractory to the stress manipulation in this 40% but not in the remaining subgroup? Were other forms of stressors tested, even preliminarily, to ascertain whether stronger or weaker stress stimuli may bring different results?

REPLY TO POINT 1: In the current study, we used social isolation as a stressor. Our choice was based on previous studies showing that social isolation can disrupt female sexual behavior. We agree with the reviewer that probably the uneven manifestation of the stress effect can be associated with the intensity of the stressor. Indeed, previous work by Jeffrey Blaustein (reviewed in; doi: 10.1016/j.jsbmb.2015.05.007) has shown that not all types of stressors can induce similar effects on female sexual behavior. Multiple stressors have been tested, including heat and light stress, food deprivation, immune challenge with the bacterial endotoxin (LPS), as well as social isolation. Although all these stressors induced quite high blood corticosterone levels during the period of exposure, only immune challenge and social isolation resulted in a decreased response to estradiol and progesterone in adulthood. A comment about this has been added to the discussion (highlighted in page 15).

POINT 2. Previous data have documented that kisspeptin output from RP3V neurones has a major role in lordosis behaviour. While according to c-fos data, these neurones are not apparently affected by the stressor, would it be possible that other parameters, such as Kiss1 mRNA expression, or more interestingly, activity patterns (as measured by fibre photometry) might have been affected, leading to impairment of activation of nNOS neurones in the VMN? This referee understands conducting fibre photometry in another set of neurones, as RP3V

Kisspeptin neurones, is not trivial, but might help to clarify the pathway. Did the authors consider this possibility?

REPLY TO POINT 2: We thank the reviewer for this comment. We were actually planning to measure the activity of the RP3V kisspeptin neurons by fibre photometry. However, it was really complicated to include such experiment in the time frame of the revisions. The position of the kisspeptin neurons makes targeting them a little bit more challenging which require some additional testing and multiple optimizations. Recording the kisspeptin neurons is on top of the TODO list of experiments, and hopefully this will be published soon in another paper.

POINT 3. In the same line, the interplay with male odours is very interesting, but the proposed pathway is unclear. Comparison between nNOS vs. Kisspeptin neuronal activity in response to odours might help to delineate which neurones are primary responsive to these stimuli and altered in response to stress.

REPLY TO POINT 3. Ideally, a suitable experiment to answer this question would be to record simultaneously, using fibre photometry, the activity of RP3V kisspeptin neurons and VMHvl nNOS, and measure the lag between the activation of the two populations. However, there is increasing evidence that are in favor of the kisspeptin neurons being upstream of the nNOS neurons. Fos data in our study showed that both kisspeptin neurons and nNOS neurons are activated by male olfactory cues. In addition, pharmacological studies (Hellier et al, 2018) have shown that kisspeptin administration in nNOS-KO females have no effect on lordosis behavior, while administration of SNAP in Kiss-KO females stimulated lordosis. Also, additional unpublished data from our lab show that RP3V kisspeptin neurons send projections towards the VHMvl nNOS neurons and make synaptic connection with them. Together, these data suggest that nNOS neurons are a downstream target of kisspeptin neurons.

POINT 4. Were the effects of peripubertal vs. adult stress compared regarding the proposed pathway? In order to define differences in the plasticity of the circuits, it might be important to check whether similar responses are obtained or not after similar stress protocols applied in adulthood.

REPLY TO POINT 4. We did not compare the effect of social isolation during puberty to adulthood. However, we think that social isolation during adulthood has no effect on female sexual behavior. In fact, in all studies from our lab and other labs, adult female mice are housed individually for weeks to test for sexual behavior, especially when they are implanted with fibres, cannulas or osmotic pumps and no negative effect was ever noticed/observed on their sexual behavior.

POINT 5. While the paper is focused on behaviours, did the authors consider the possibility to assess the impact of the stress protocol on reproductive hormonal profiles, such as endogenous progesterone and LH levels? This might be relevant from a mechanistic perspective, since an impairment of ovulatory function may lead to defective progesterone secretion (e.g., no or lower number of corpora lutea in stressed animals). In other words, while replacement experiments suggest that a central component is in place, the apparent progesterone dependence suggests that perturbations of the neuroendocrine profiles caused by pubertal stress might contribute to the observed phenomena.

REPLY TO POINT 5. We have added new data on the estradiol levels during stress exposure (Fig. 2a) and adulthood (Fig. 2b), as well as the concentration of progesterone during adulthood (Fig. 2c). We are aware that more data will be needed to fully characterize the reproductive hormonal profile. However, this was not the main scope of the current study.

POINT 6. Pharmacological experiments with the NO donor are interesting. Did the authors consider the possibility to compare SNAP results with those of kisspeptin administration in control vs. stressed animals? No matter what the results are (if SNAP and kisspeptin do the same or not), this might be informative, especially considering that nNOS VMN neurones are responsive to kisspeptin.

REPLY TO POINT 6. New data comparing the effects of peripheral administration of kisspeptin to SNAP is added to the manuscript (Fig. 9b). We have found that in contrast to SNAP, kisspeptin administration in MR females induced a small increase in lordosis behavior which was not statistically significant.

POINT 7. In the discussion of the data, the authors tend to compare current results on impaired lordosis with low sexual desire in women. I would suggest this connection is further supported by previous literature, as low sexual desire in women may result from multiple components. Interestingly, recent studies suggest a connection between kisspeptin and sexual desire in men, so that kisspeptin treatment might enhance sexual desire in patients suffering hyposexual desire. Do the authors consider there might be some connection with present findings?

REPLY TO POINT 7. In unpublished data, we tested the same stress protocol that we used in the present study on male mice. However, we found no effect of social isolation on male sexual behavior, including sexual preference and performance. So, it is difficult to suggest a connection between hyposexual desire, pubertal stress and kisspeptin in males. Besides, the role of the rather scattered RP3V kisspeptin neuronal population is still unclear in males.

However, since both male-typical behavior and arcuate kisspeptin neurons develop during-gestation, one can hypothesize that prenatal stress might lead to reduced sexual behavior in males, by disrupting the development of the arcuate kisspeptin population. Future studies should address this.

REVIEWER COMMENTS

Reviewer #1 (Remarks to the Author):

The authors have responded to all my queries adequately.

Reviewer #2 (Remarks to the Author):

I have reviewed the revised version of the manuscript and found that most of my concerns have been fully addressed, with new data now complementing the manuscript. I would like to congratulate the authors for this beautiful work. I have only few points to make and a couple of minor suggestions, upon which I strongly support the publication of the manuscript.

Line 104-105: The authors did now provide information on the vaginal opening, that as they correctly point out is the first external sign of sexual maturation. Yet vaginal opening doesn't correspond to puberty onset in mice (in contrast to what seen in rats). Hence, a lack of difference in the vaginal opening (that indeed would be unlikely) doesn't exclude a delay of puberty onset, which is expected to occur around P45, i.e. around the end of the isolation protocol (and thus quite possibly affected by that protocol). If the authors have monitored the first estrous, i.e. puberty onset, it would be great to add this info to the manuscript.

Lines 259-264: In lines 259-262 authors mention that OVX+E2 animals showed lower nNOS activation to male urine compared to OVX+E2+P4 and that it was similar to what observed for the OVX+P4 group. However, in the sentence lines 262-264 authors say that there is no difference between OVX+P4 and OVX+E2+P4. Could authors rephrase (do they refer to female urine instead of male ?) and also add the p values (asterisks, ns) in their supplementary figure S6 (or in the figure legend)?

Line 260: check figure references (3i, 4r)

Lines 286-288: Authors conclude that nNOS activation is modulated by progesterone but not estradiol. Eventually however, as the authors point out in their discussion lines 427-430 is not progesterone, but rather the presence of both ovarian steroids (this is also what figure S6 seems to demonstrate). Could the authors clarify/ rephrase this part?

Line 316 : Reference to figure 9b instead of 6b.

Line 328-330 : I would suggest the authors to precise that it is a peripheral administration of SNAP/ vehicle (since they also performed a central infusion).

Line 331-338 : Authors add this very elegant experiment and measure the levels of the phosphorylation of nNOS neurons in the VMH in the activation site Ser1412 upon vehicle or SNAP injection. Their new results show that the number of nNOS neurons being phosphorylated is unaltered between control and MR females in both vehicle and SNAP injected groups, as well as that SNAP injection efficiently activated the nNOS enzyme in both control and MR females. It would be informative for authors to provide the nNOS/pnNOS ratio, complementary to the pnNOS-ir numbers. This is extremely interesting particularly because the nNOS-ir neuronal numbers are decreased in MR females.

Line 396-398: I would suggest the authors to modify their statement and say that SNAP treatment resulted in "similar number of activated nNOS cells" instead of a "similar activation of nNOS neurons

Figure 9b,d : Could authors add p values/ ns in the figures ?

Reviewer #3 (Remarks to the Author):

The authors have extensive revised their work in response to the comments of the referees. As a result, the work has been improved and the conclusions have been strengthened. The authors are commended for their hard work in revising their manuscript.

Please find below our corresponding replies to reviewer 2. Most comments have been addressed and changes were made accordingly (highlighted in green in the revised manuscript).

REVIEWER COMMENTS

Replies to Reviewer #2:

Reviewer #2 has raised multiple minor suggestions that have been considered.

I have reviewed the revised version of the manuscript and found that most of my concerns have been fully addressed, with new data now complementing the manuscript. I would like to congratulate the authors for this beautiful work. I have only few points to make and a couple of minor suggestions, upon which I strongly support the publication of the manuscript.

Line 104-105: The authors did now provide information on the vaginal opening, that as they correctly point out is the first external sign of sexual maturation. Yet vaginal opening doesn't correspond to puberty onset in mice (in contrast to what seen in rats). Hence, a lack of difference in the vaginal opening (that indeed would be unlikely) doesn't exclude a delay of puberty onset, which is expected to occur around P45, i.e. around the end of the isolation protocol (and thus quite possibly affected by that protocol). If the authors have monitored the first estrous, i.e. puberty onset, it would be great to add this info to the manuscript.

⇒ **Indeed, we agree with the reviewer that a lack of difference in vaginal opening does not exclude a delay of puberty onset. However, we decided not to monitor the onset of the first estrous cycle because even though it is to be expected around P45 as rightly pointed out by the reviewer, we would have started to monitor our subjects around P40 until at least P50 (the end of the isolation protocol). We were worried that it would induce additional stress in our subjects.**

Lines 259-264: In lines 259-262 authors mention that OVX+E2 animals showed lower nNOS activation to male urine compared to OVX+E2+P4 and that it was similar to what observed for the OVX+P4 group. However, in the sentence lines 262-264 authors say that there is no difference between OVX+P4 and OVX+E2+P4. Could authors rephrase (do they refer to female urine instead of male?) and also add the p values (asterisks, ns) in their supplementary figure S6 (or in the figure legend)?

Line 260: check figure references (3i, 4r).

⇒ **We thank the reviewer for pointing this out. We were actually referring to female urine. The sentence has been rephrased, the p values have been added, and the figure references have been revised. The modifications are highlighted on page 11.**

Lines 286-288: Authors conclude that nNOS activation is modulated by progesterone but not estradiol. Eventually however, as the authors point out in their discussion lines 427-430 is not progesterone, but rather the presence of both ovarian steroids (this is also what figure S6 seems to demonstrate). Could the authors clarify/ rephrase this part?

⇒ **We agree with the reviewer. The sentence highlighted on page 12 has been corrected now.**

Line 316: Reference to figure 9b instead of 6b.

⇒ **Reference to the figure has been corrected (highlighted on page 13).**

Line 328-330: I would suggest the authors to precise that it is a peripheral administration of SNAP/ vehicle (since they also performed a central infusion).

⇒ **Indeed, we performed peripheral injections of SNAP/vehicle to measure the phosphorylation of the nNOS neurons. This information is now added to the manuscript (highlighted on page 14).**

Line 331-338: Authors add this very elegant experiment and measure the levels of the phosphorylation of nNOS neurons in the VMH in the activation site Ser1412 upon vehicle or SNAP injection. Their new results show that the number of nNOS neurons being phosphorylated is unaltered between control and MR females in both vehicle and SNAP injected groups, as well as that SNAP injection efficiently activated the nNOS enzyme in both control and MR females. It would be informative for authors to provide the nNOS/pnNOS ratio, complementary to the pnNOS-ir numbers. This is extremely interesting particularly because the nNOS-ir neuronal numbers are decreased in MR females.

⇒ **We thank the reviewer for suggesting this very interesting experiment. Indeed, the nNOS/pnNOS ratio can add interesting information about the effect of SNAP treatment on the reduced nNOS numbers in MR females. It will also be interesting to determine whether the effect is transient or permanent. Unfortunately, we do not have any brain tissue left to perform this additional staining. Therefore, to address this point, we would have to start this experiment from scratch which would take us months to finish, since it will include breeding the animals, exposing them to pubertal stress, verifying the behavior, collecting their brains, and performing and quantifying the staining. This will not be possible within the proposed time frame of the current revision. However, in our follow-up studies, we will include this interesting experiment.**

Line 396-398: I would suggest the authors to modify their statement and say that SNAP treatment resulted in “similar number of activated nNOS cells” instead of a “similar activation of nNOS neurons.

⇒ **The sentence has been rephrased (page 17).**

Figure 9b,d : Could authors add p values/ ns in the figures ?

⇒ **p values and ns have been added to the figures 9b and 9d.**

REVIEWERS' COMMENTS

Reviewer #2 (Remarks to the Author):

I would like to congratulate the authors for addressing the queries effectively and for their diligence that has greatly enhanced the quality of their work. I have no further comments to add, and I wholeheartedly recommend the publication of their manuscript.